# Comparison of GRUAN Data Products for Meisei iMS-100 and Vaisala RS92 Radiosondes at Tateno, Japan

Shunsuke Hoshino[1], Takuji Sugidachi[2], Kensaku Shimizu[2], Eriko Kobayashi[3], Masatomo Fujiwara[4], and Masami Iwabuchi[3]

[1]Office of Numerical Prediction Modeling Fundamental Technology, Numerical Prediction Division, Information Infrastructure Department, Japan Meteorological Agency, 1-2 Nagamine, Tsukuba, Ibaraki, 305-0052, Japan
[2]Meisei Electric Co., Ltd., 2223 Naganumamachi, Isesaki, Gunma, 372-8585, Japan
[3]Aerological Observatory, 1-2 Nagamine, Tsukuba, Ibaraki, 305-0052, Japan
[4]Faculty of Environmental Earth Science, Hokkaido University, Kita 10 Nishi 5, Kita-ku, Sapporo, 060-0810, Japan

**Correspondence:** Shunsuke Hoshino (shoshino@met.kishou.go.jp)

**Abstract.** A total of 99 dual soundings with Meisei iMS-100 radiosonde and Vaisala RS92 radiosondes were carried out at Aerological Observatory of the Japan Meteorological Agency, known as Tateno (36.06 °N, 140.13 °E, 25.2 m; WMO station number 47646), from September 2017 to January 2020. Global Climate Observing System (GCOS) Reference Upper-Air Network (GRUAN) data products (GDP) from both sets of radiosonde data for 59 flights were subsequently created using a documented processing program along with the provision of optimal estimates for measurement uncertainty. Differences in radiosonde performance were then quantified using these GDPs. For daytime observations, the iMS-100 temperature is around 0.5 K cooler than RS92-GDP in the stratosphere with significant differences in the upper troposphere and lower stratosphere in consideration of combined uncertainties. For nighttime observations, the difference is around -0.1 K and data are mostly in agreement. For relative humidity (RH), iMS-100 is around $1 - 2$ %RH higher in the troposphere and 1 %RH smaller in the stratosphere than RS92, but both GDPs are in agreement for most of the profile. The mean pressure difference is $\leq 0.1$ hPa, the wind speed difference is from -0.04 $\mathrm{m\,s^{-1}}$ to +0.14 $\mathrm{m\,s^{-1}}$, the wind direction difference $\leq 6.4$ °, and the root mean square of vector difference (RMSVD) for wind is $\leq 1.04$ $\mathrm{m\,s^{-1}}$.

## 1 Introduction

The Aerological Observatory of the Japan Meteorological Agency (JMA), called as 'Tateno' (location: 36.06° N, 140.13° E, 25.2m above mean sea level), was established in 1920 and has played a leading role in the operation of all JMA radiosonde stations. The Tateno station was chosen as a candidate site for the Global Climate Observation System (GCOS) Reference Upper-Air Network (GRUAN; Seidel et al., 2009; Bodeker et al., 2016) in 2009, and was certificated as a GRUAN site in 2018. The Vaisala RS92-SGP radiosonde (referred to here as RS92; Dirksen et al., 2014) was used for routine observation at Tateno site from December 2009. While RS92-SGP provides the data with high time resolution, that is, high altitude resolution, and highly reliable data and has GDP, it was required to seek an alternative radiosonde model for operational reasons. This is because the payloads often fall within the greater Tokyo metropolitan region (i.e., highly populated area) and therefore for

safety reasons use of lighter instrumentation (the weight of RS92 is 290 g) is necessary. In response to this request, the Meisei RS-11G radiosonde (Kizu et al., 2018; Kobayashi et al., 2019; Hoshino et al., 2022a), whose weight is 85 g, was released in 2012, and started to be used at Tateno in July 2013. The GDP for RS-11G was developed and certificated in 2019. Following the

25 release of the Meisei iMS-100 radiosonde (Kizu et al., 2018; Hoshino et al., 2022a), smaller ($55 \times 53 \times 131$ mm$^3$) and lighter (40 g) model, in 2014, it has been used since September 2017. The temperature and humidity sensor of iMS-100 are basically identical with RS-11G, but relative humidity sensor of iMS-100 has a dedicated thermometer and GPS module is different between the two. RS-11G has also been used in operation at Syowa station (Antarctica) since March 2018, while iMS-100 has been used at Minamitorishima and other JMA stations since 2017, as well as at stations of other meteorological service

providers and numerous research institutes and universities. Syowa and Minamitorishima are currently candidate GRUAN sites.

Figure 1 gives an overview of the GRUAN data stream. Raw data output from the ground system ("dc3cb" or "mwx" files for RS92 radiosonde data and "JFMT files" for iMS-100 radiosonde data) and meta-data files are submitted to GRUAN Lead Centre (LC) for archiving, and GRUAN LC issues a data processing ticket file including the data and meta-data file names, IDs

for the measurement and the product, the output file name, and other elements for processing. The data processing center (PC; Lindenberg for RS92 and Tateno for iMS-100 radiosondes), collects raw data, meta-data and ticket files from GRUAN LC and performs processing to create GDP files in NetCDF format. The output GDP file are submitted to GRUAN LC. The GDP files are distributed at National Centers for Environmental Information (NCEI; ftp://ftp.ncdc.noaa.gov/pub/data/gruan/processing/level2/) after quality checking.

The evaluation of GDP for RS-11G (RS-11G-GDP.1; Kizu et al., 2019) using GDP for RS92 (RS92-GDP.2; Sommer et al., 2012) in dual flight was discussed in Kobayashi et al. (2019). In Kobayashi et al. (2019), the RS-11G-GDP.1 temperature is around -0.4 K lower than RS92-GDP.2 in daytime measurement in stratosphere, while nighttime measurements generally agree well. The RS-11G-GDP.1 RH was 2 %RH smaller than the RS92-GDP.2 for 90 %RH – 100 %RH, and the RS-11G-GDP.1 was around 5 %RH larger than the RS92-GDP.2 at values lower than 50 %RH, so RH difference exceeds 2 %RH between 500

and 150 hPa in both daytime and nighttime data. The pressure difference was 0.5 hPa in the troposphere, and the geopotential height difference was around 10 – 20 m in the stratosphere.

Although the sensors of iMS-100 are almost identical with RS-11G, the data processing algorithms have some distinct updates for IMS-100-GDP, such as the correction of hysteresis effect of the RH sensor and the correction of geoid model used in GPS module.

In this paper, Sect. 2 describes the instrumentation used and GRUAN data products (including a brief description of data processing) for iMS-100 and RS92. Sect. 3 outlines dual soundings and Sect. 4 describes the comparison analysis method. The comparison results are given in Sect. 5, and Sect. 6 discusses outcomes from a dual flight of iMS-100 and a chilled-mirror hygrometer, Meisei SKYDEW (Sugidachi, 2019). Sect. 7 summarizes the findings. The related abbreviations are listed in Appendix A.

## 2 Instrumentation

### 2.1 Sensor material and specifications

Table 1 shows the specifications of iMS-100 and RS92. More detailed specification for each model are described in Vaisala Oyj. (2013), Dirksen et al. (2014), Meisei Electric Co., Ltd. (2020) and Hoshino et al. (2022a). The ground system for iMS-100 is Meisei MGPS2, and that for RS92 was Vaisala DigiCORA III and was replaced to Vaisala MW41 in September 2019.

### 2.2 GRUAN data processing for iMS-100

Figure 2 gives an overview of GRUAN data processing for iMS-100. The ticket file (*.gpt) contains the raw data file name (*JFMT.DAT) and meta data file name (*.gmd), the individual identification number for observation in the GRUAN data archive, and the file name for processed data. GMDB is the abbreviation for the "GRUAN Meta DataBase", which contains information on payload equipments (such as radiosondes, balloons, parachutes, unwinders, rigs). In pre-processing, the lag of data acquisition timing in the processor between temperature / humidity and GPS related data (time and positioning) are adjusted, relative humidity is re-calibrated with additional ground check data (at the 0 %RH and 100 %RH conditions, where available), geometric altitude is corrected with the finer geoid model (see Section 2.2.3), and initial data for ascending are extracted by identifying the start and end of ascent. Usually, the end of ascent is when the radiosonde reached its maximum altitude (thus, the start of descent or the end of radiosonde signals), but ascent data are truncated to ensure its reliability in the event of missing temperature values (i.e., gaps in observation), over a period of 180 seconds or more. Initial data are processed at each step as outlined below to produce the final data. Derived data related to humidity (such as partial water vapor pressure, frost point temperature, and precipitable water vapor) are also calculated. To support climate record quality, GDP data include their uncertainty at each measurement point. The processed data are stored in NetCDF format, and the GDP for iMS-100 radiosonde data is noted as "IMS-100-GDP". The processing algorithm is documented in Hoshino et al. (2022a) for detail, so outline and difference from RS-11G-GDP.1 (Kizu et al., 2018; Kobayashi et al., 2019) is described in this paper.

### 2.2.1 Temperature

An overview of temperature calculation is given in Fig. 3. $T_0$, $U_0$, $P_{surf}$ and $H_{fin}$ are uncorrected temperature values, uncorrected relative humidity values, surface pressure and corrected geopotential height (see Sect.2.2.3). If the length of the string or the unwinder is equal to or less than 10 m, temperature spike filtering is applied when pressure is below 30 hPa. For this filtering and radiation correction based on the heat balance equation (JMA, 1995), provisional pressure $P_0$ is derived using $T_0$, $U_0$, surface pressure data, $P_{surf}$, and $H_{fin}$ (Section 2.2.3). $T_1$, the temperature after spike filtering is $T_0$ if no filtering is applied. The ascent rate, $asc$, at the previous sampling time is used as the ventilation speed, $vent$, for calculation of the radiation correction. The ventilation speed at the release time is set as $0 \, \mathrm{m \, s^{-1}}$. The moving averages of $T_1$ within $\pm 1$ second are calculated to determine the uncorrected temperature data with radiation correction, $T_2$. The amount of radiation correction, $T_{rad\_corr}$ is calculated using $T_2$, $P_0$, $P_{surf}$, $vent$, and solar elevation angle which is derived from GPS time, $\phi_0$ (latitude), and

$\lambda_0$ (longitude). $T_{rad\_corr}$ is subtracted from $T_2$, to determine $T_3$. As the thermistor is not completely spherical, there is a small dependence on the angle to sunlight. Accordingly, the Kaiser filter is applied to $T_3$ to minimize this orientation effect to obtain the final temperature value, $T_{fin}$. The uncertainty budget for temperature is shown in Table 2. The uncertainties are classified as either correlated or uncorrelated when the source of the uncertainty is systematic or random, respectively (Dirksen et al., 2014).

For example, the uncertainties due to the smoothing processes are classified as uncorrelated uncertainties, and uncertainties due to calibration or solar radiation correction are considered correlated uncertainties. Figs. 4 (a) and (b) show the vertical profiles of temperature and its uncertainty at 00 UTC on 15 September, 2017. The dashed line in Fig. 4 (a) indicates the tropopause height. The blue, green, and red lines in Fig. 4 (b) are correlated, uncorrelated, and total uncertainty, respectively. The blue and red lines almost overlap, but the correlated uncertainty is dominant. For daytime observation, the correlated uncertainty increases with altitude because the amount of solar radiation correction increases, and the uncertainty from it also increases.

### 2.2.2  Relative humidity (RH)

An overview of RH calculation is given in Fig. 5. $T_{hum}$ and $L$ are temperature of humidity sensor and length of unwinder or string, respectively. For the time-lag correction, which corrects the effect of the time constant delay, the response time coefficients are constant for the RS-11G-GDP.1, but are dependent on whether the RH sensor is absorbing or desorbing water vapor molecules (i.e., whether RH is increasing or decreasing, respectively) for the IMS-100-GDP. The trend of RH change is estimated from the slope of the tangent of $U_0$ using the method proposed by Savitzky and Golay (1964) using the $U_0$ values within $\pm 5$ seconds values. The response time, $\tau_{hum}$, is calculated from obtained trend value together with the temperature of the RH sensor, $T_{hum}$. Time-lag correction is applied to $U_0$ to determine $U_1$ values, which are separated into low and high frequency components, $U_{1low}$ and $U_{1high}$, via the digital filtering with a cut-off period set as four times the period of pendulum motion, $T_{pend}$. High frequency components sometimes exhibit spike-like variations, which are considered to be associated with water droplets or ice attached on the humidity sensor. These spike-like variations are removed by using a minimum-pass filter, moving averages and an infinite impulse response (IIR) filter to determine $U_{2high}$. The low and high frequency components, $U_{1low}$ and $U_{2high}$ are summed to create $U_2$. For the RS-11G-GDP.1, the hysteresis errors, that is, the small error remaining after the time-lag correction possibly due to the inhomogeneity of change in the thin-film polymer RH sensor (termed the "slow-regime" in Dupont et al., 2020), are not corrected but only considered as an uncertainty component. For the IMS-100-GDP, the estimation of the hysteresis errors are formulated as follows: the slow part ratio and its response times for absorption and for desorption are assumed as 3%, 300 seconds and 12000 seconds, respectively. The RH of the slow part, $U_{delay}$, and the hysteresis corrected RH, $U_3$ are derived via iterative calculation with:

$$U_{delay}(t) = \frac{U_2(t) + (1-\alpha)\tau_{hys}U_{delay}(t-1)}{(1-\alpha)\tau_{hys}+1} \tag{1}$$

$$U_3(t) = (1+\tau_{hys})U_{delay}(t) - \tau_{hys}U_{delay}(t-1) \tag{2}$$

where $\alpha$ is the slow part ratio, $\tau_{hys}$ is the response time for the slow part and $U_3$ is the corrected RH.

The temperature-humidity-dependence (TUD) correction is applied to minimize RH biases at lower temperatures in the same way as for RS-11G-GDP.1 by Eqs.3 and 4, but the coefficients are redetermined via additional chamber experiments and flight comparison results with Cryogenic Frostpoint Hygrometer (Vömel et al., 2007, 2016) (Table 3).

$$\Delta U = (K_{0,0} + K_{0,1} \times U_3 + K_{0,2} \times U_3^2)(1 - (K_{1,0} + K_{1,1} \times U_3)) \exp(-T_{hum}/(K_{2,0} + K_{2,1} \times U_3))$$
$$+ (K_{3,0} + K_{3,1} \times U_3 + K_{3,2} \times U_3^2) \tag{3}$$

$$U_4 = U_3 - \Delta U \tag{4}$$

During soundings, the temperature of the RH sensor is not necessarily equal to the ambient air temperature because of solar heating and the thermal lag of the RH sensor. Values monitored with an RH sensor that are higher (lower) than those of the
ambient air would result in dry (wet) biases. Accordingly, $U_4$ values need to be translated to RH with respect to saturation vapor pressure associated with air temperature using the following equation (referred as "$T_s/T_a$ correction").

$$U_{fin} = U_4 \times \frac{e_s(T_{hum})}{e_s(T_a)} \tag{5}$$

where $e_s(T)$ is saturated water vapor pressure over liquid water, based on the formulation by Hyland and Wexler (1983):

$$\log e_s = \frac{a_0}{T} + a_1 + a_2 T + a_3 T^2 + a_4 T^3 + a_5 T^4 + a_6 \log T, \tag{6}$$

where $T$ is the temperature [K] with the coefficients, $a_i$, as shown in Table 4.

$U_{fin}$ is the final value of RH. The uncertainty budget for RH is shown in Table 5. The uncertainties due to calibration and time-lag correction are classified as correlated uncertainties. The uncertainty due to low-pass filter is uncorrelated uncertainty. The uncertainty due to the IIR filter has correlated components propagating from the time-lag correction and uncorrelated components from the low-pass filter. The uncertainties from the hysteresis correction, the TUD correction, and the $T_s/T_a$
correction are calculated by propagation from the uncertainties from all previous steps. The final correlated uncertainty is obtained as synthesizing the calibration uncertainty and the correlated uncertainty after the $T_s/T_a$ correction is applied. Figs. 4 (c) and (d) show the vertical profiles of RH and its uncertainty at 00 UTC on 15 September, 2017. The blue, green, and red lines in Fig. 4 (d) are correlated, uncorrelated, and total uncertainty, respectively. The green and red lines almost overlap at most altitudes. This means the uncorrelated uncertainty is dominant. The correlated uncertainty has some magnitude (1 – 4
140    %RH) in the troposphere but are nearly negligible above 50 hPa level. If $U_{fin} + u(U)$ is less than 0 %RH, these are treated as missing values (NaN) in output.

### 2.2.3 Geopotential height

An overview of geopotential height calculation flow is given in Fig. 6. The raw geometric altitude, $Z_0$, is calculated from ellipsoidal height derived with GPS signal and the internal geoid model of iMS-100, but the grid resolution of the internal geoid model with $10° \times 10°$ is too coarse and the interpolated geoid height may differ from the actual geoid height by $20\,\mathrm{m}$ or more in some regions (near Japan is the one of the region with large differences. see Fig. 7). Accordingly, geometric altitude is recalculated using the geoid model (Pavlis et al., 2012) with a finer grid ($5' \times 5'$) to derive $Z_1$. The results of the verification using the GNSS simulator show that $Z_0$ was found to have a $\simeq 1\,\mathrm{s}$ delay with respect to the assumed altitude. $Z_1$ is corrected for this delay to obtain $Z_2$. Also this verification shows that there is a delay of several seconds in measurements just after launch. This delay is attributed to Kalman filtering in positioning by the GPS module and the difference becomes negligible several seconds after launch. Accordingly, the altitude in this "transition" period is interpolated with the known release altitude and the observed altitude at the end of transition to obtain $Z_3$, which is then applied with a moving average to determine the final geometric altitude, $Z_{fin}$. The geopotential height, $H_{fin}$ is calculated using:

$$H_{fin} = \frac{g_\phi}{g_0} \frac{R \times Z_{fin}}{R + Z_{fin}} \tag{7}$$

where $R$ is the radius of the earth (6378136.0 m), $g_0$ is standard gravity acceleration ($9.80665\,\mathrm{m\,s^{-2}}$), and $g_\phi$ is gravity acceleration at the observation latitude.

The geopotential height uncertainty consists of the components due to vertical positioning precision (correlated) and smoothing (uncorrelated).

### 2.2.4 Pressure

The iMS-100 radiosonde is not equipped with a pressure sensor. In recent years, radiosondes without a pressure sensor are widely used. This is in part because of reduction of weight and manufacturing costs. Also, the accuracy of the atmospheric pressure derived using geopotential height obtained from GPS measurements and hydrostatic equation-based approximation is sufficient in the troposphere and superior in the stratosphere (e.g., Nash et al., 2011).

Pressure is calculated from the geopotential height, $H_{fin}$ (2.2.3), temperature, $T_{fin}$, and RH, $U_{fin}$, using the hydrostatic equation.

$$P_i = P_{i-1} \exp\left[ -\frac{g_0}{T_{vm} R_d} \Delta H \right] \tag{8}$$

where $P_i$ is the calculated pressure [hPa], $P_{i-1}$ is the pressure at the previous level (or time step, i.e., 1 second before), $\Delta H$ is the thickness [m] between the two consecutive levels, $\Delta H = H_i - H_{i-1}$, $g_0$ is standard gravity acceleration, $T_{vm}$ is the

average virtual temperature between the two consecutive levels [K], and $R_d$ is the specific gas constant of dry air. The virtual temperature is calculated as

$$T_v = T\frac{(1 + r_v/\varepsilon)}{1 + r_v} \tag{9}$$

with

$$r_v = \frac{\varepsilon e}{P - e} \tag{10}$$

where $T_v$ is the virtual temperature [K], $r_v$ is the mixing ratio of water vapor, $\varepsilon$ is the ratio of molecular weight between water vapor and dry air ($= 0.622$), $P$ is pressure, $e$ is water vapor pressure calculated from temperature and RH based on the water vapor pressure equation by Hyland and Wexler (1983). For $P$, $P_{i-1}$ is used. . The pressure uncertainty is calculated at each data point based on the error propagation rule from temperature, RH, and geopotential height uncertainties from the previous time step. The propagation for correlated and uncorrelated uncertainties is calculated separately, and combined to derive the total uncertainty. An example of the pressure uncertainty profile is shown in Fig.4 (e).

## 2.2.5 Wind

An overview of wind calculation flow is given in Fig. 8. While the initial (unsmoothed) wind speed and direction are derived from GPS Doppler shift for RS-11G-GDP.1, the initial wind speed $wspeed_0$ and direction $wdir_0$ are derived as motion vectors from longitude ($\lambda$; $lon_0$ in Fig. 8) and latitude ($\phi$; $lat_0$ in Fig. 8) based on GPS positioning for IMS-100-GDP. The great-circular distance, $d$, and direction, $\theta$, between the position at $t_i$ ($\lambda_i$, $\phi_i$) and $t_{i+1}$ ($\lambda_{i+1}$, $\phi_{i+1}$) are given with spherical trigonometry by:

$$d = R\arccos(\sin\phi_i \sin\phi_{i+1} + \cos\lambda_i \cos\lambda_{i+1} \cos(\lambda_{i+1} - \lambda_i)) \tag{11}$$

$$\theta = 90 - \arctan(\frac{\cos\lambda_i \tan\lambda_{i+1} - \sin\phi_i \cos(\lambda_{i+1} - \lambda_i)}{\sin(\lambda_{i+1} - \lambda_i)}) \tag{12}$$

where $R$ is the radius of the earth, and $d$ and $\theta$ are rendered as $wspeed_0$ and $wdir_0$, respectively.Smoothing process is same as RS-11G-GDP.1. The wind vectors with a wind speed of $wspeed_0$ and a wind direction of $wdir_0$ are decomposed to the zonal and meridional wind components, $u_0$ and $v_0$. Each of these components are smoothed by a low-pass filter using Kaiser window with a cut-off frequency of $f_{cutoff} = 1/T_{pend}$, which $T_{pend}$ is the cycle of pendulumn motion by string with length $L$. to derive the final wind components, $u_{fin}$ and $v_{fin}$, respectively. The final wind components are synthesized to wind speed, $wspeed_{fin}$, and direction, $wdir_{fin}$.

Uncertainties of zonal and meridional wind components consist of that related to the positioning uncertainty and that from smoothing by the Kaiser filter. Both components are uncorrelated uncertainties. The positioning component is estimated as 2.5 m/s, which is statistically derived using the GPS simulator. The uncertainties of wind speed and direction are calculated from the partial differentiation of the formula that converts wind components to wind speed or direction.

## 2.3 GRUAN data processing for RS92

As data processing for RS92-GDP version 2 has detailed by Dirksen et al. (2014), only a brief description is provided here.

Raw temperature data are corrected for solar radiation and heat spike errors. Solar radiation errors relate to overall direct and scattered solar irradiance, ambient pressure, and ventilation, and are estimated at the GRUAN Lead Centre from a radiative transfer model that takes into account the solar elevation angle at the time of monitoring. Vaisala radiation error correction data are also available in table form. GRUAN data processing for RS92 involves application of the average of the two, as it remains unclear which of the two correction models is more appropriate. Heat spike errors are removed via a low-pass digital filter with a cut-off frequency of 0.1 Hz.

RS92 RH sensors have a temperature-dependent dry bias. GRUAN data processing corrects for this based on multiplication with an empirical correction factor before other forms of the correction are applied. Raw RH data are corrected for radiation dry bias, sensor time lag, and temperature-dependence errors. Radiation dry bias is caused by solar heating on the RH sensors, and the same approach as for the temperature sensor is used to estimate the amount of the correction required. The RH sensor response slows at low temperatures, and time lag becomes significant below -40 °C. This is corrected based on the relationship between a time constant and temperature using a low-pass filter in the GRUAN data product for RS92 (Dirksen et al., 2014).

The RS92 used at Tateno has a pressure sensor and a GPS receiver, both of which can be used to calculate geopotential height. Pressure measurement data are used to derive geopotential height in the lower part of the profile where the signal-to-noise performance of the pressure sensor is sufficiently good, and measurements from the GPS sensor are used in the upper part of the profile. The altitude of the switch is typically between 9 and 17 km (Sommer et al., 2016). The pressure sensor is recalibrated against the reference value from a station barometer during the ground check, and calculation is performed to determine the correction factor for application to the entire pressure profile during sounding (Dirksen et al., 2014). U and V data are retrieved from the Doppler shift in the GPS carrier signal, and noise is removed using a low-pass digital filter. The smoothed data are converted into wind speed and direction values (Dirksen et al., 2014). Uncertainties of each parameters for RS92 are described in Dirksen et al. (2014).

While the authors used version 2 of the RS92 GDP, version 3 is supposed to be available in the near future (Sommer et al., 2016) and it would be useful to redo the analysis with it.

## 3 Method used for dual sounding

Dual soundings with iMS-100 and RS92 for intercomparison have been conducted from September 2017 to January 2020 at 00 UTC (09 LT; daytime) or 12 UTC (21 LT; nighttime) once a week, except for wind conditions in which a payload may fall to populated areas around metropolitan Tokyo (in general, from July to mid-September with relatively weak westerly winds in the upper troposphere due to northward displacement of the subtropical jet stream and the stratospheric easterly winds). There were 99 flights during this period (52 daytime, 47 nighttime).

Payload configuration for dual sounding is shown in Figure 9. A 1200 g balloon was used for all dual sounding. The iMS-100 and RS92 radiosondes were hung on both ends of a 0.9 – 1 m corrugated plastic board or bamboo rod, and this rig was covered with aluminum tape to reduce the effects of radiation based on the proposal of Rohden et al. (2016).

## 4 Method for comparison

GDPs data both for iMS-100 and RS92 are collected at 1 s intervals. Temporally simultaneous measurements were compared using the two statistical approaches adopted by Kobayashi et al. (2019) to evaluate differences in the data products. The first method, which is the layer mean in the ensemble of dual flights (described in Section 4.4), is to obtain a profile of differences between the two data products. This is necessary to create a homogenized data set for the climatological discussion at a site where there were instrument changes. The second method, which is the consistency verification using the uncertainty per single measurement level following Type B evaluation in Immler et al. (2010) (described in Section 4.5), is necessary to know if the results obtained in the first method can be regarded as being significantly different.

### 4.1 Screening with quality assessment

Prior to statistical comparison, irregular or inappropriate data for comparison should be excluded. RS92-GDP, processed by the GRUAN LC, has been checked with a quality assessment algorithm by the GRUAN LC. As the algorithm for IMS-100-GDP in this study is designed with reference to that for RS92-GDP, the algorithm for RS92-GDP is described before that for IMS-100-GDP.

For RS92-GDP, the first quality screening is based on differences between radiosonde and references sensor in the ground check. The thresholds are 1.5 hPa for pressure, 1.0 K for temperature and 1.5 %RH for RH (Sommer, 2013). Data with larger difference are not used. The second screening is based on the quantified uncertainty estimates (Sommer, 2013). This screening is based on the idea that data with uncertainties exceeding the criteria are of questionable reliability and need to be verified individually. The thresholds for uncertainty, $u_{\mathrm{spec}}^{RS92}$, are calculated as:

$$u_{\mathrm{limit}}^{\mathrm{RS92}}(T) = 0.5 \tag{13}$$

$$u_{\mathrm{limit}}^{\mathrm{RS92}}(U) = 0.025U + 2.5 \tag{14}$$

$$u_{\mathrm{limit}}^{\mathrm{RS92}}(P) = 0.0004P + 0.6. \tag{15}$$

When uncertainties for 95 % or more of data in the entire profile are within $u_{\mathrm{limit}}$, the profile is approved and otherwise labeled as "Checked".

For RH, however, the criterion in Eq.14 is too strict, because it is lower than actual typical values especially for 50 %RH – 70 %RH. Thus, the original formula as per Eq.16 is used here.

$$u_{\text{limit}}^{\text{RS92}}(U) = -0.000578U^2 + 0.0925U + 1.457 \tag{16}$$

However, the lower limit of $u_{\text{limit}}^{\text{RS92}}(U)$ is set to 2.5 %RH. Figs. 4 (a) – (c) show the profiles of temperature, RH and $u_{(}U)$
at 00 UTC on 09 November, 2018. The blue and red line represent the criteria calculated by Eqs.14 and 16, respectively. Figs. 4 (d) and (e) show whether the data are within (blue) or exceeding (red) the criteria by Eqs.14 and 16 at each altitude, respectively. Figs. 10 (f) and (g) show the percentage of data within or exceeding in the whole profile, thus, being equivalent to the rearrangement of Figs.10 (d) and (e). The percentage of data within the criteria using Eqs.14 and 16 are 90 % and 98 % in Figs.10 (d) and (e), respectively. Therefore, this case is classified as "Checked" with the LC's criteria (i.e., 95 %). However,
it is included in the analysis in this study.

For IMS-100-GDP, the first screening is performed at ground check under room conditions in preparation. The thresholds of differences from reference sensors for temperature and RH are $\pm$ 0.5 °C and 7 %RH, respectively. Sensors values exceeding these criteria are not used for observation.

The second screening checks contamination or changes in the RH sensor specifications associated with icing. In some
270 soundings with iMS-100, abnormal RH profiles such as the one shown in Fig. 11 (for 12UTC on 22 September, 2017) are sometimes observed. For RS92, there is little need to consider potential freezing of its humidity sensor since it is heated. On the other hand, the very slow RH decreasing after passage a through supercooled layer (red shaded in Fig. 11(b)) and relatively high RH values in the stratosphere for IMS-100-GDP are not explained by the hysteresis, but probably due to contamination and changes in the RH sensor specifications related to icing or freezing during passage through supercooled droplet clouds. As
checking to determine whether the radiosonde has passed through such clouds is impractical from the RH profile, ice saturated regions (pink shaded layer in Fig. 11(b)) are considered. However, as not all data associated with ice-saturated clouds passage are contaminated, the length of the ice-supersaturated region (ISSR) and the RH and water vapor mixing ratio at the top of the upper troposphere and lower stratosphere (UTLS) are used for screening in this study. ISSR determination is based on saturated water vapor for liquid water and ice calculated using Hyland and Wexler equation (Hyland and Wexler, 1983). The observed
RH is the ratio of water vapor pressure to $e_{\text{s\_liq}}$, and is limited to $U_{\text{s\_ice}} = e_{\text{sat\_ice}}/e_{\text{sat\_liq}}$ for the ice phase. Accordingly, ISSR is the layer in which $U > U_{\text{sat\_ice}}$. In this study, two adjacent ISSRs with intervals of $\leq 60$ seconds or where the minimum of $U - U_{\text{sat\_ice}}$ is more than -10 %RH (i.e., with no dry layer in the interval) are merged and treated as a single ISSR.

For screening of ice-contaminated profiles, the probability of icing, $Pr_{\text{ice}}$, is derived using logistic regression analysis (e.g., Cox, 1958) after variable selection from the length of ISSR, RH and the volume mixing ratio at several levels. The 452 routine
observations (twice per day) with a single payload taken from April to November 2018 at Tateno are used as training data. The Python scikit-learn package (Pedregosa et al., 2011) was used for the actual coefficient derivation. The $Pr_{\text{ice}}$ is calculated with:

$$Pr_{ice} = \frac{1}{1 + \exp(-22.38 + 7.18 \times 10^{-3}T_{\text{ISSR\_max}} + 0.517U_{\text{ST1}} + 0.977U_{\text{ST2}} + 0.105W_{\text{ST1}} + 0.427W_{\text{ST2}})} \tag{17}$$

where subscripts ST1 and ST2 represent the data points at 2000 m and 4000 m above the tropopause in geopotential height, $U$ [%RH] is RH, $W$ [ppmv] is the water vapor volume mixing ratio, and $T_{\text{ISSR\_max}}$ is the pass-through period of maximum ISSR in seconds. In this study, profiles with $Pr_{\text{ice}} > 0.5$ are considered as potentially ice-contaminated data and excluded in the RH analysis below.

The third screening is based on uncertainties for temperature, RH and pressure as with RS92. The thresholds for these values are derived from Eqs.18 – 20, respectively. Coefficients are determined empirically.

$$u_{\text{limit}}^{\text{iMS}}(T) = 0.7 \tag{18}$$

$$u_{\text{limit}}^{\text{iMS}}(U) = \begin{cases} -1.438 \times 10^{-6}U^3 + 9.867 \times 10^{-4}U^2 - 1.020 \times 10^{-2}U + 3.081 \text{(for daytime)} \\ -1.195 \times 10^{-5}U^3 + 2.631 \times 10^{-3}U^2 - 7.308 \times 10^{-2}U + 3.351 \text{(for nighttime)} \end{cases} \tag{19}$$

However, the lower limit of $u_{\text{limit}}^{\text{iMS}}(U)$ is set to 3.3 %RH for both cases.

$$u_{\text{limit}}^{\text{iMS}}(P) = 1.609 \times 10^{-9}P^3 - 4.589 \times 10^{-6}P^2 + 4.120 \times 10^{-3}P + 3.810 \times 10^{-2}. \tag{20}$$

For example, the criteria for temperature and RH uncertainty are shown as black lines in Figs. 4 (b) and (d), respectively. In this case, the RH uncertainty exceeds the criteria near 370 hPa and 250 hPa. The ratio criterion is set to 90 %, thus, if more than 90% the whole profile has data whose uncertainty does not exceed the threshold, the IMS-100-GDP is included in the following analysis. For the case in Fig. 4, the ratio of data for which the uncertainty is within the threshold is greater than 90 %. On the other hand, Fig.12 (at 00 UTC on 6 December, 2019) shows the rejected case with only 84 % of the data within the RH criteria due to the large uncertainty above 15 hPa level. For the case in Fig.13 (at 00 UTC on 24 November, 2017), the large pressure uncertainty layer exceeding the criteria is found between 540 hPa and 230 hPa. Only 82 % of the data are within the pressure criteria, and thus this case is rejected. In this case, the large correlated uncertainty in the middle troposphere is caused by large VDOP, i.e., the precision of GPS positioning (not shown). It should be noted that this quality assessment and screening are for this study only, and are not authorized as the standard for GDPs. The standard method for quality assessment of GDPs is currently under discussion by the quality task force in the GRUAN community.

As a result of the screening process described above, the 59 dual sounding flights (29 daytime and 30 nighttime) shown in the Table 6) are used as a suitable data set for comparison. Seasonal profiles for temperature, RH, and wind speed are shown in Figs. 15 – 17, with seasonal classification (MA, MJJ, ASON and DJF) described in Sect. 4.4. The major factors associated with data screening is illustrated in Fig. 14 for each season and daytime/nighttime. The rejection rates are low in DJF but high in MA and MJJ. The primary reasons are the large uncertainties in humidity or pressure, icing of humidity sensor for iMS-100, and the bad RS92 sensors, i.e., humidity differences exceeding thresholds to the reference sensor in the ground check. Thus, the data are rejected mainly due to humidity sensor issues. It maybe is related to the climatology at Tateno, where it tends to be dry in winter and humid and highly variable with altitude in MA and MJJ (Fig. 16). The highly variable humidity may cause large uncertainties for time-lag correction or hysteresis correction. Passing through ISSRs increases the likelihood of the

humidity sensor icing. For RS92, the ground check of the humidity sensor is performed with the assumption that the humidity of the chamber filled with molecular sieves is 0 %RH, but the molecular sieves in chamber gradually becomes damp because it takes in water vapor from the atmosphere. At Tateno, molecular sieves were replaced when the humidity difference between RS92's sensors exceeded 2 %RH, which may have exceeded the RS92-GDP screening criteria.

## 4.2 Timestamp adjustment

Radiosonde observation have timestamps from the relevant sounding system (for iMS-100, this is based on received GPS clock data). As there may be minor discrepancies in balloon-launch time stamps, these data are time-adjusted using the temperature profile. In this study, shift registration in functional data (Ramsay and Silverman, 2005) is used for adjustment using the temperature data from between 5 and 20 minutes of the launch time of RS92. Temperature data for each radiosonde, $T_i(t)$, in this period are converted to functional data, $x_i(t)$. Shift values, $\delta_i$, are calculated to minimize the registered curves sum of squares errors (REGSEE), which is defined as

$$\text{REGSEE} = \sum_i \int \left[x_i(t + \delta_i) - \hat{\mu}(t)\right]^2 dt, \tag{21}$$

where $\hat{\mu}(t)$ is the mean function of $x_i(t)$. For calculation of $\delta_i$, the scikit-fda package (Carreño, 2020) for python is used and actual adjustment values (seconds to shift) are derived with

$$\delta t = \delta t_{\text{RS92}} - \delta t_{\text{iMS}}. \tag{22}$$

When $\delta t > 0$, timestamps for iMS-100 are shifted backward by $|\delta t|$, and vice versa.

## 4.3 Data pretreatment

Profiles with $> 10\,\%$ of abnormal data points as whole profile are excluded via screening as described in Section 4.1, while abnormal data are also seen at individual points in overall normal profiles (e.g, superadiabatic lapse rates). Such data points should be excluded from statistical comparison. In this study, superadiabatic lapse rate layers and abnormal wind data immediately before balloon burst are pre-treated for masking.

## 4.4 Statistical comparison for binned layers based on pressure

After timestamp adjustment, per-second differences between iMS-100 and RS92 measurements were calculated and differences were allocated to the 13 pressure layers based on RS92 pressure data ($P_i^{\text{RS92}}$, where $i$ indicates the time step) based on Kobayashi et al. (2012, 2019). The bins for the 13 layers are listed in Table 7.

$A_{i,j}^{\text{iMS}}$ and $A_{i,j}^{\text{RS92}}$ are values at the time step $i$ for iMS-100 and RS92 in the $j$th dual sounding data set ($j = 1, \ldots, M$; $M$ is the number of data sets, here 59), respectively. The differences between the two radiosonde types, $\Delta A_{i,j} = A_{i,j}^{\text{iMS}} - A_{i,j}^{\text{RS92}}$, are averaged for each pressure layer with:

$$\overline{\Delta A_{k,j}} = \frac{\sum_{i=i_{b,j}}^{i_{t,j}} \Delta A_{i,j}}{N_{k,j}} \tag{23}$$

where $k$ is the layer number ($k = 1, \ldots, 13$), $N_{k,j}$ is the number of data points in the layer $k$, $i_{b,j}$ and $i_{t,j}$ are the bottom and top time step at the $j$th data set, respectively. The ensemble mean of the difference in each pressure layer is calculated as

$$\langle \Delta A_k \rangle = \frac{\sum_{j=1}^{M} \overline{\Delta A_{k,j}}}{M}. \tag{24}$$

The ensemble standard deviation of the mean difference for individual pressure layers is

$$\sigma(A_k) = \sqrt{\frac{\sum_{j=1}^{M} (\overline{\Delta A_{k,j}} - \langle \Delta A_k \rangle)^2}{M}} \tag{25}$$

Comparison of wind data is performed with the indices of the wind speed bias, $BIAS$, and the root mean square error of the vector difference, $RMSVD$ (CGMS, 2003) for each pressure layer at each dual sounding. $BIAS$, vector difference ($VD$), mean vector difference ($MVD$), standard deviation of vector difference ($SD$) and $RMSVD$ are calculated as

$$BIAS_{k,j} = \sum \frac{1}{N} (V_{i,j}^{iMS} - V_{i,j}^{RS92}) \tag{26}$$

$$VD_{i,j} = \sqrt{(u_{i,j}^{iMS} - u_{i,j}^{RS92})^2 + (v_{i,j}^{iMS} - v_{i,j}^{RS92})^2} \tag{27}$$

$$MVD_{k,j} = \frac{1}{N_{k,j}} \sum VD_{i,j} \tag{28}$$

$$SD_{k,j} = \sqrt{\frac{1}{N_{k,j}} \sum (VD_{i,j} - MVD_{k,j})^2} \tag{29}$$

$$RMSVD_{k,j} = \sqrt{MVD_{k,j}^2 + SD_{k,j}^2}, \tag{30}$$

where $V$, $u$, and $v$ is the wind speed, zonal wind speed and meridional wind speed, respectively. The mean difference of wind direction (Szantai et al., 2007), $\overline{\Delta DIR}$, is also used.

$$\overline{\Delta DIR_{k,j}} = \frac{180}{N\pi} \sum \arccos\left(\frac{\boldsymbol{V}_{i,j}^{iMS} \cdot \boldsymbol{V}_{i,j}^{RS92}}{V_{i,j}^{iMS} V_{i,j}^{RS92}}\right) \tag{31}$$

where $\boldsymbol{V}$ is the wind vector. The ensemble mean of these parameters is then calculated as for Eq.24.

Statistics for each pressure layer are calculated for daytime, nighttime, and individual seasons. Due to the safety consideration described in Section 3, few dual soundings are performed in July and August, and the weather conditions for August flights are categorized as for autumn rather than summer. Thus, in this study, the flights from August to November are categorized as those for autumn (here, ASON), and previous studies for seasonal comparison campaigns of radiosondes at Tateno (Kobayashi, 2015; Kobayashi and Hoshino, 2018), the "summer" covered the period from May to June, flights in March and April are categorized with spring (here, MA), those from May to July are categorized with summer (MJJ), and those from December to February are categorized with winter (DJF).

### 4.5 Method for verification of consistency with uncertainties

Uncertainty estimation for RS92 and iMS-100 GDPs are described in Dirksen et al. (2014) and Hoshino et al. (2022a), respectively.

Immler et al. (2010) proposed terminology for comparing pairs of independent measurements of the same quantity for consistency using estimated uncertainties as described in the following: Consider two independent measurements, $m_1$ and $m_2$, of the same measurand with standard uncertainties, $u_1$ and $u_2$, respectively. Under the assumptions that the measurements $m_1$ and $m_2$ have independent errors with measurement uncertainties $u_1$ and $u_2$, and the difference $m_1 - m_2$ has a normal distribution, the probability that

$$|m_1 - m_2| > k\sqrt{u_1^2 + u_2^2} \tag{32}$$

occurs only by chance, is roughly 4.5% for $k = 2$ and 0.27% for $k = 3$. If Eq. 32 is true for $k = 2$, it is very likely that the two measurements did in fact not measure the same thing, probably due to an unrecognized or unaccounted-for systematic effect in one or both measurements. Immler et al. (2010) proposed an expression for the degree of consistency as shown in Table 8. This approach is Type B evaluation of uncertainty. For Type A evaluation, Immler et al. (2010) concluded that it is not expected to play an important role within GRUAN, and thus it is not considered in this study.

For statistical consistency check, the total consistency ranks shown in Table 8 ($k < 1$: consistent, $1 \le k < 2$: in agreement, $2 \le k < 3$: significantly different, or $k \ge 3$: inconsistent) between RS92 and iMS-100 within a specific pressure layer for a particular parameter are estimated as the 95 % percentile value of consistency ranking numbers within the layer.

## 5 Results

### 5.1 Temperature

Figure 18 shows the ensemble mean (lines) and standard deviation (error bar) of temperature differences, for (a) daytime and nighttime for all season, (b) seasonal daytime, and (c) seasonal nighttime. In the stratosphere, IMS-100-GDP value are around -0.5 K lower than RS92-GDP for the daytime. For the nighttime, differences are around -0.1 K below the 10 hPa level. Seasonal differences are small.

Figure 19 shows the percentage of consistency rank in each layer for (a) daytime and (b) nighttime. The percentages of significantly different or inconsistent data exceed 50 % in L07 ($100 - 70$ hPa), 30 % in L08 ($70 - 50$ hPa) and L09 ($50 - 30$ hPa) for daytime observation (Fig 19(a)). However, 80 % of data are in agreement for nighttime comparison (Fig 19(b)). Fig. 20 shows the distribution of temperature at L09 ($50 - 30$ hPa), L07 ($100 - 70$ hPa), L05 ($200 - 150$ hPa) and L03 ($500 - 300$ hPa). Fig. 20 (a1), (a2) and (a3) show differences in L09, L07 and L05 in general are normally distributed with a sufficient sample size. Accordingly, daytime temperature differences in the stratosphere and the upper troposphere are associated with systematic effects. These differences (especially between 100 and 30 hPa) are attributed to difference in the solar radiation correction models. Further discussion about the contributions of different radiation heating correction methods to the tem-

perature difference needs other observation data like satellites or other types of radiosonde, like RS41. But GNSS-RO-based temperature data is very limited and no comparative observations have been made at Tateno between three sondes (iMS-100, RS92, and RS41). Therefore, additional discussion is expected after the results of comparisons between iMS-100 vs RS41 and RS92 vs RS41 are published. However, some samples show significant differences ($> \pm 0.5$K) even in the troposphere or for nighttime soundings (not shown), which are associated either with issues during flights or calibration problems.

Kobayashi et al. (2019) found that the RS-11G-GDP.1 temperature data are about -0.4 K lower than RS92-GDP.2 data for daytime observations in the stratosphere. This means that the IMS-100-GDP temperatures show larger differences from RS92-GDP.2 than RS-11G-GDP.1 temperatures do. On the other hand, the ratio of data that are evaluated as "consistent" or "in agreement" with RS92-GDP.2 temperatures is greater for IMS-100-GDP than RS-11G-GDP.1. This is probably due to the newly included correction for the sensor orientation effects in IMS-100-GDP, which will increase the uncorrelated uncertainty. Further investigation is needed using intercomparison results between RS-11G and iMS-100 radiosondes.

### 5.2 Relative humidity

Figure 21 shows the ensemble mean (lines) and standard deviation (error bars) of RH differences, for (a) daytime and nighttime for all seasons, (b) seasonal daytime and (c) seasonal nighttime. This shows that iMS-100 RH is around $1 - 2$ %RH larger than RS92 RH around the tropopause and -1 %RH smaller in the stratosphere. Unlike temperature, systematic differences (i.e.,biases) between daytime and nighttime soundings are small but seasonal variations are large in the upper troposphere and the lower stratosphere. Figure 22 shows the ensemble mean of RH differences for six RH ranks. In the lower troposphere (below 500 hPa level), differences are below 2 %RH and exhibit limited correspondence with RH values. In the middle and upper troposphere (500 – 100 hPa), the difference increases with altitude for 10 – 90 %RH. For data with RH $\leq$ 10 %RH, the difference is within $\pm$ 1 %RH for layers above 500 hPa level; iMS-100 RH is wetter in the troposphere and drier in the stratosphere. For data with RH $>$ 90 %RH, the dataset is limited above 300 hPa and drier than RS92 figures for the middle troposphere (500 – 300 hPa).

Kobayashi et al. (2019) found that RS-11G-GDP.1 RH data shows about 2 %RH dry tendencies for conditions with RH $>$ 90 %RH and about 1 %RH wet tendencies for conditions with RH $\leq$ 10 %RH. These different behaviors between the RS-11G-GDP.1 and IMS-100-GDP with respect to RS92-GDP.2 in very dry and very wet conditions make the $\Delta U$ profiles different in the lower troposphere (below 700 hPa) and in the stratosphere (above 50 hPa) (See Fig.21 in this study and Fig. 11 of Kobayashi et al., 2019). In particular, the 1 %RH wet bias of RS-11G-GDP.1 with respect to RS92-GDP.2 and the dry bias of IMS-100-GDP in the stratosphere are a notable differences. The major reason of these differences could be the including of hysteresis correction in the IMS-100-GDP. The ratio of data that are evaluated as "consistent" or "in agreement" with RS92-GDP.2 RH data in the troposphere is greater for RS-11G-GDP.1. This is also due to the inclusion of hysteresis correction in the IMS-100-GDP, because the RH profiles in the troposphere often show rapid changes as shown in Fig.16. Further investigation on the differences between RS-11G and iMS-100 results by making intercomparison flights is a future task.

Figure 23 shows the percentage of consistency rank in each layer for (a) daytime and (b) nighttime. In the troposphere and lower stratosphere (L1 – L7, below 70 hPa level), around 10 – 20 % of data are significantly different or inconsistent. The RH

profiles for individual flights (Fig. 16) show that RH often shows rapid changes. For the flight at 12 UTC (21 LT) on 19 April, 2019 (Fig. 24), iMS-100 shows a slow tendency in relation to the rapid decrease in RH (e.g., at about 330 hPa), compared to RS92. This difference in response to rapid changes, especially in case from high to low humidity, is considered a reason for the inconsistency of 1-second RH values between the two radiosondes. As described in Sect. 2.2.2, the iMS-100's RH sensor has hysteresis with the large time constant, but RS92's RH sensor is heated and its hysteresis is negligible. This difference in characteristics of RH sensor could cause the large difference, especially in rapid decreasing RH case. In the stratosphere (L8 – L13, above the 70 hPa level), RH data from iMS-100 and RS92 seem to be almost in agreement.

## 5.3 Pressure

Figure 25 shows the ensemble mean (lines) and standard deviation (error bars) of pressure differences for (a) daytime and nighttime for all season, (b) seasonal daytime, and (c) seasonal nighttime. The absolute value of ensemble mean difference is less than 0.4 hPa, but there are cases with large differences in the lower troposphere (below 700 hPa level). This may be attributable to the effect of pressure differences between RS92 pressure sensor and the barometer used for surface observation. The pressure of IMS-100, with no pressure sensor, is derived from recursive calculation via the hydrostatic equation, so that the surface pressure is equal to that observed using ground-based barometer. Meanwhile, RS92 involves independent pressure sensor usage, meaning that near-surface pressure may differ between GDPs. A histogram of RS92 GDP surface pressure error for ground-based barometer content (Fig. 26) shows that the difference is not normally distributed around zero. RS92 pressure tends to be slightly higher than ground-based barometer content with a difference median of 0.33 hPa, which is greater than barometer uncertainty (0.06 hPa for $k = 1$). The consistency check for surface pressure between the barometer and RS92 resulted in "significantly different" or "inconsistent" in 20 cases. The effect of this difference decreases with height, but is more noticeable near the ground.

For RS-11G-GDP.1, the pressure is about -0.5 hPa lower for daytime observation in middle of troposphere, but the difference between IMS-100-GDP and RS92-GDP.2 is little in those layer statistically. At this time, the reason of this difference of pressure is not clear.

## 5.4 Geopotential height

Figure 27 shows the ensemble mean (lines) and standard deviation (error bars) of geopotential height differences, for (a) daytime and nighttime for all seasons, (b) seasonal daytime, and (c) seasonal nighttime. The difference in geopotential height is around $2 – 3$ m in the lower and middle troposphere, but becomes larger with altitude above 100 hPa level and become about 10 m at 20 hPa. This tendency is attributed to difference in geoid height as referenced by IMS-100-GDP and RS92-GDP. As described in Section 2.2.3, the grid size of the original geoid model used for the iMS-100 GPS module is $10\,^\circ \times 10\,^\circ$, which is replaced with a $5\,' \times 5\,'$ model for geometric height calculation in GDP processing. The grid size difference causes significant discrepancy in geoid and geometric height especially for the northwest Pacific basin around Japan. Figures 28(a) and (b) show geoid height for IMS-100-GDP and original iMS-100, respectively, 28(c) shows differences in geoid height and typical radiosonde track for each season (green for MA, red for MJJ, orange for ASON and blue for DJF), and 28(d) shows

geoid correction values for typical seasonal tracks. The difference increases with height for all seasons. The grid size of the geoid model for RS92-GDP is unknown but as geometric height values are used without modification, geoid model differences may have caused geopotential height differences.

The geopotential height difference between IMS-100-GDP and RS92-GDP.2 seems to have no noticeable difference with that between RS-11G-GDP.1 and RS92-GDP.2, although the RS-11G-GDP.1 is not corrected geoid model. This implies that
the geoid model used in RS-11G has enough resolution for GDP.

## 5.5 Wind

Figure 29 compares wind for (a) wind speed bias, (b) RMSVD and (c) wind direction difference for all seasons. The difference is small enough with BIAS from -0.04 to +0.14 $\mathrm{m\,s^{-1}}$, RMSVD is less than 1.04 $\mathrm{m\,s^{-1}}$ except for L13 (above 10 hPa), and $\overline{\Delta DIR}$ is less than 6.4 °.
Simlarly to the verification for temperature and relative humidity, the consistency checks in each pressure-binned layer for wind speed and direction are conducted (see Fig. 30). The figure shows that more than 20% of the data for wind speed are "significantly different" or "inconsistent" in the lowest layer (below 700 hPa level) and between 100 hPa and 20 hPa levels. For wind direction, more than 20% of the data in most layers and more than 50% of the data in the lowest layer and between 100 hPa and 20 hPa levels are "significantly different" or "inconsistent." This result appears to be inconsistent with the verification
result using mean differences for each layer mentioned above.

To understand this difference, the wind speed difference and the uncertainty profile at 12 UTC on July 12, 2019, are shown in Fig. 31 as an example. The inset figures are an enlarged view of the L07 (100 – 70 hPa), where the wind speed was estimated as "significantly different." In Fig. 31a, which shows the wind speed profile, the overall change trends are consistent for both GDPs, but the RS92-GDP (the orange line) looks smoother than the IMS-100-GDP (the blue line). This difference in
smoothing causes the variability of wind speed differences (black line in Fig. 31b), resulting in some of the data exceeding the range of the synthetic uncertainty thredshold (the green line in Fig. 31b for $k = 2$). As a result, the ratio of data that is estimated as "consistent" or "in agreement" is less than 95 % in the layer, and the consistency check gives "significantly different" or "inconsistent." Although the two GDPs are not in agreement in the layer, the differences due to this smoothing are canceled out when averaged and therefore do not appear as statistical errors. Similar "significant difference" problems due to smoothing
is found in the wind direction profiles. Fig. 32 shows the wind direction difference and the uncertainty profile for the same sounding, and the inset figures show the enlarged view of the L02 (700 – 500 hPa). Fig. 32 suggests that the significant difference in wind direction caused by the difference in the degree of smoothing method as with the wind speed. Additionally, for the lowest level (below 700 hPa level, especially in the boundary layer), the differences in positioning performance of GPS modules immediately after launch or the possible unstable posture of the payload due to the double pendulum motion, rig
rotation, or swaying may have caused a slight difference of wind calculation.

## 6 Comparison with a frost-point hygrometer

Due to the technical limitations of the RH sensors mounted on operational radiosondes in low temperature and dry conditions, GRUAN requires comparisons of RH data with values from reference instruments. At Tateno, the comparison flights with radiosondes and cryogenic frostpoint hygrometer (CFH; Vömel et al., 2007, 2016) have been conducted twice a year since 2015. However, since the R-23 (HFC-23) liquid cryogen material used to cool the mirror of CFH was regulated under the Montreal Protocol, a new frost-point hygrometer, Meisei SKYDEW (Sugidachi, 2019) was adopted in 2020 for cooling the mirror with a Peltier element instead. Figure 33 shows results from the comparison conducted at 06 UTC (15 LT) on 21 October 2020. Although the differences in RH are significant (with $\Delta U$ exceeding the extended uncertainty ($k = 2$)) around the tropopause (92.2 hPa, 16983.0 m, and -73.4 °C) and in the lower part of the troposphere (800 – 400 hPa), more than 80 % of IMS-100-GDP RH data except for L02 (700 – 500 hPa) and L07 (100 – 70 hPa) are "consistent" or "in agreement". In particular, almost all data above 70 hPa are consistent with SKYDEW. In this study, the uncertainty of SKYDEW RH is not implemented in the consistency check. If the uncertainty of SKYDEW RH is estimated, the result is expected to be more consistent between RH of IMS-100-GDP and SKYDEW. Fig. 33(c) shows that the RH of IMS-100-GDP and SKYDEW are generally in agreement for the entire profile and consistent for most of the troposphere.

## 7 Summary

To characterize GDPs for iMS-100 and RS92, data from dual soundings conducted at Tateno from September 2017 to January 2020 are analyzed in this study. The iMS-100 temperature is around 0.5 K lower than RS92-GDP for daytime observation in the stratosphere and over 50 % of data from between 100 and 70 hPa and over 30 % from between 70 hPa and 30 hPa shows significant differences from RS92-GDP. For nighttime observation, the difference is around -0.1 K with over 80 % of data showing in agreement both in the troposphere and the stratosphere. The difference for daytime measurements in the stratosphere is attributed to the correction procedures for solar radiation heating and differences in sensor characteristics.

The iMS-100 RH is around 1 – 2 %RH higher in the troposphere and 1 %RH lower in the stratosphere than RS92, but both GDPs are generally in agreement in the troposphere and stratosphere. The difference may be larger in places where rapid RH change occurs. A comparison flight with the SKYDEW, frost-point hygrometer, shows that iMS-100 RH agrees well with SKYDEW both in the troposphere and stratosphere.

While there are some cases where with significant differences for pressure are observed in the lower troposphere ($\geq 700$ hPa) in the consistency check, the mean pressure difference is less than 0.4 hPa. The difference in geopotential height is around 2 – 3 m in the lower and middle troposphere, but increases with altitude above 100 hPa level, from 10 m at 20 hPa. This relationship between height and related differences may stem from differences in the geoid model used for the two GDPs.

In wind comparison, although the consistency check based on uncertainty may estimate the wind speed and direction for both GDPs as significantly different for each simultaneous data, the comparison parameters show good correspondence; BIAS is between -0.04 and +0.14 $\mathrm{m\,s^{-1}}$, RMSVD is lower than 1.04 $\mathrm{m\,s^{-1}}$ except for above 10 hPa, and $\overline{\Delta DIR}$ is smaller than 6.4 °

in the statistical comparison for each pressure-binned layer. This seemingly contradictory result is mainly due to the difference in the degree of smoothing for the two GDPs.

The modified data processing described in Sect. 2.2.1 – 2.2.2 will be implemented to processing for RS-11G to create new version of RS-11G-GDP, which will be evaluated with intercomparison campaigns written in Kobayashi et al. (2019) (RS-11G and RS92) and Kobayashi and Hoshino (2018) (iMS-100 and RS-11G). Further direct comparison between RS-11G-GDP and IMS-100-GDP will be discussed in the future study.

    This study involved evaluation of the characteristics of IMS-100-GDP values with RS92-GDP as a reference, as the latter
is certified as a GRUAN data product. GRUAN certification for iMS-100 is underway, and ongoing analysis of GDP data is considered important for the provision of high-quality products to the climate research/monitoring community. Since February 2020, regular dual soundings of iMS-100 and Vaisala RS41 (the successor to RS92), have been ongoing. RS41-GDP is under development (Rohden et al., 2021) and the comparison results will be published when sufficient IMS-100-GDP and RS41-GDP data available. In addition, IMS-100-GDP, RS41-GDP and M10-GDP (Dupont et al., 2020) are also candidates of the reference
data in the WMO 2022 Upper-Air Instrument Intercomparison Campaign (CIMO Task Team on Upper-air Intercomparison, 2020) which will be conducted in 2022. These data will support further evaluation and improvement of IMS-100-GDP.

    The interpolation and the estimation of uncertainty for data missing periods are discussed in some articles. For example, Fassò et al. (2020) proposed a method for temperature data using the Gaussian process, and Colombo and Fassò (2022) attempted to apply it to RH data. These studies will be considered for future improved versions of the IMS-100-GDP.

*Data availability.* The GRUAN data products for IMS-100-GDP.2 are available from https://doi.org/10.5676/GRUAN/IMS-100-GDP.2 (Hoshino et al., 2022b). The GRUAN data products for RS92-GDP.2 are available from https://doi.org/10.5676/GRUAN/RS92-GDP.2 (Sommer et al., 2012).

## Appendix A: Abbreviations

**EPS**  Expanded Polystyrene Styrofoam

**GBAS**  Ground-Based Augmentation System

**GCOS**  Global Climate Observing System

**GDP**  GRUAN Data Product

**GMDB**  GRUAN Meta DataBase

**GNSS**  Global Navigation Satellite System

**GPS**  Global Positioning System

**GRUAN**  GCOS Reference Upper Air Network

**ISSR**  Ice-supersaturated region

**JMA**  Japan Meteorological Agency

**JFMT**  JMA transmission Format

**LC**  Lead Centre

**Meisei**  Meisei Electric Co., LTD

**NCEI**  NOAA National Centers for Environmental Information

**RH**  Relative humidity

**RMSVD**  Root mean square of vector difference

**SBAS**  Satellite-Based Augmentation System

**WMO**  World Meteorological Organization

*Author contributions.*  SH, TS, KS, and EK developed the iMS-100 GRUAN data product, with SH and EK performing data analysis and creating the figures. All authors provided ideas and contributed to interpretation of the results. The first draft of the paper was written by SH, with all authors contributing to improvement of the paper.

*Acknowledgements.*  The authors are grateful to the Aerological Observatory, the Observation Division of JMA's Atmosphere and Ocean Department, and Meisei Electric for their support and helpful advice. Thanks are also due to GRUAN Lead Centre staff for their support.

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

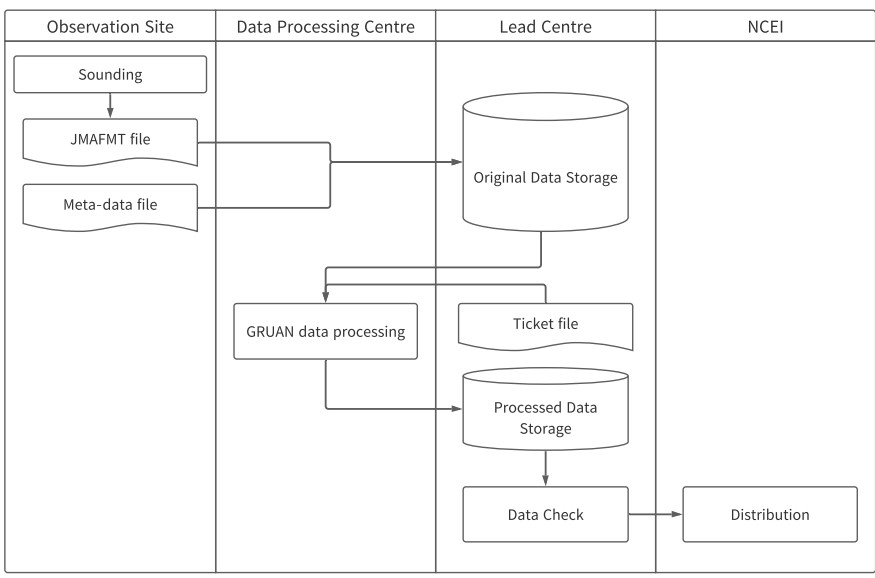

**Figure 1.** GRUAN data files stream

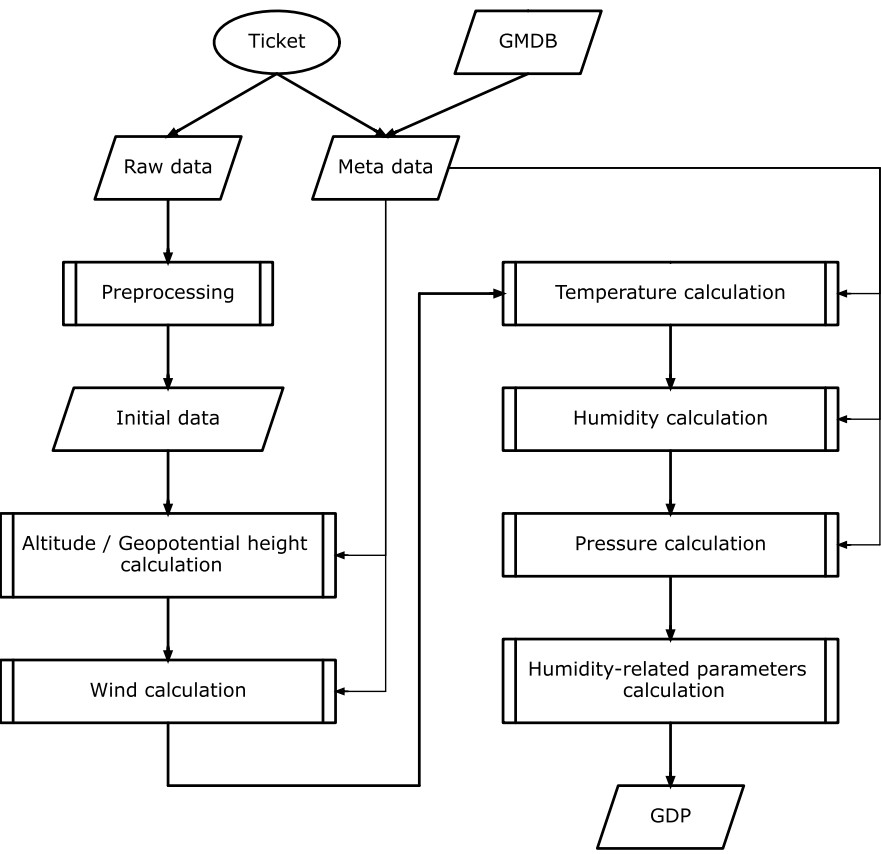

**Figure 2.** GRUAN data processing for iMS-100 (GMDB: GRUAN Meta Database). Cicle is the process starting point. Parallelograms and rectangles with double lines on the left and right sides represent data and subprocesses for calculation, respectively.

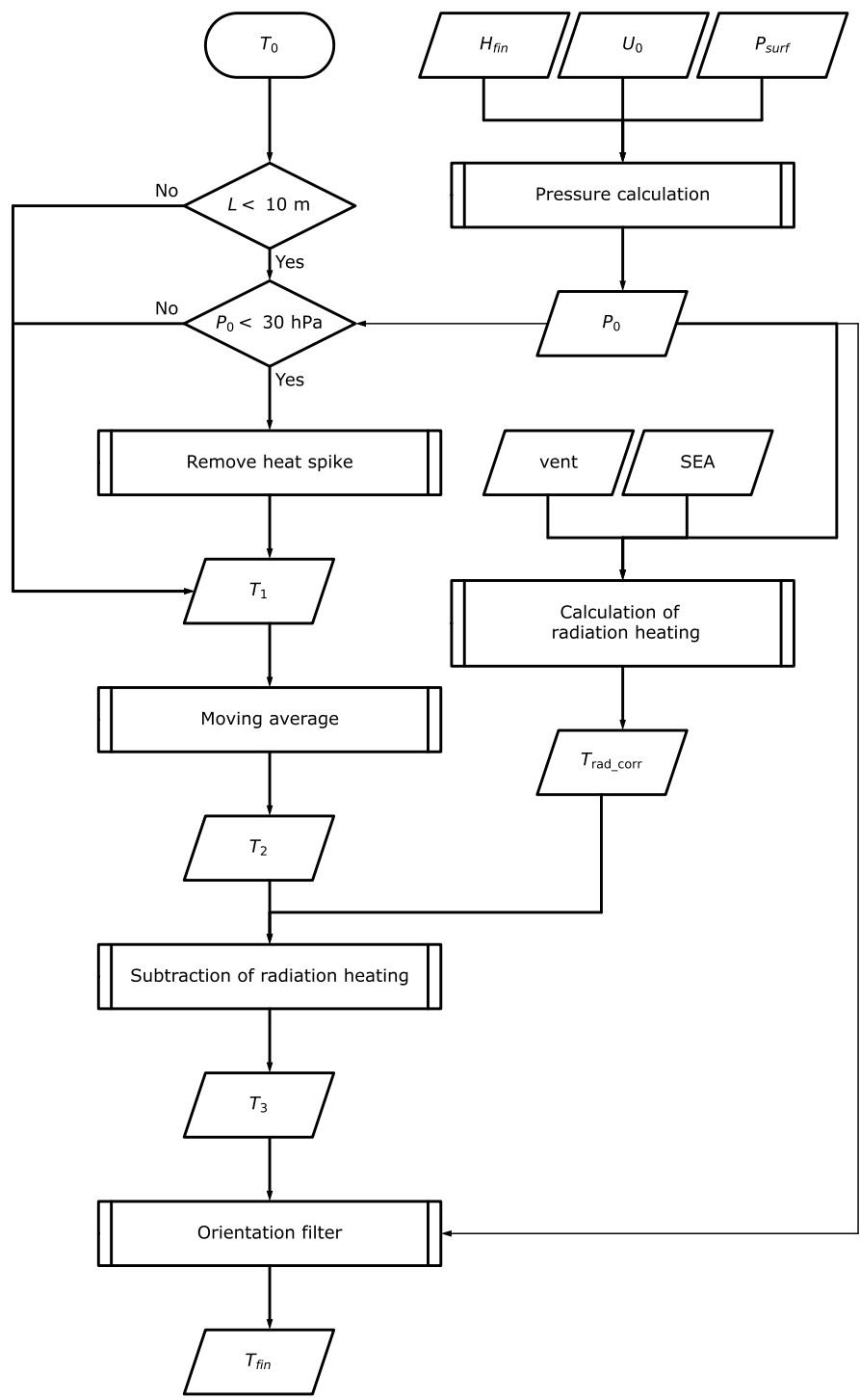

**Figure 3.** Temperature processing for iMS-100 (L: the total string (or unwinder) length; SEA: sun elevation angle derived from time and radiosonde position). Meanings of shapes are same as Fig. 2. Diamonds represent decision.

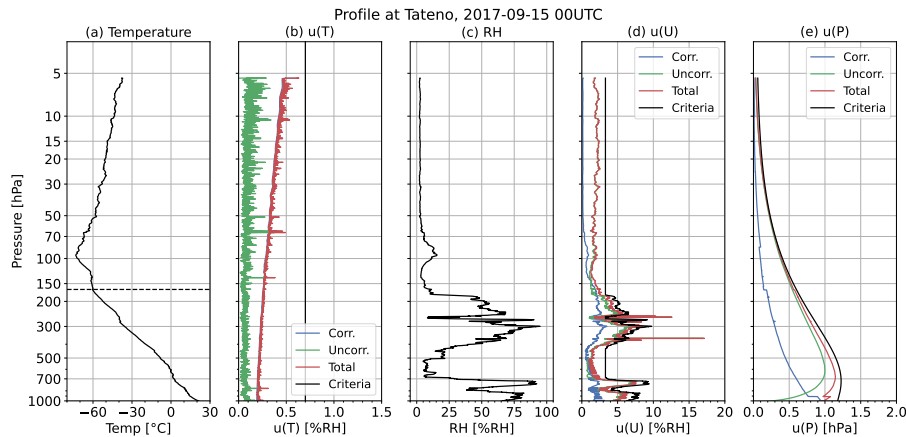

**Figure 4.** The profile of (a)temperature, (b) temperature uncertainty, (c) RH, (d) RH uncertainty, and (e) pressure uncertainty at 00 UTC on 15 September, 2017. The dashed line in (a) is the tropopause height. The blue, green, and red lines in (b), (d), and (e) are correlated, uncorrelated, and total uncertainty, respectively. Black lines in (b), (d), and (e) show the criteria for screening described in Sect. 4.1.

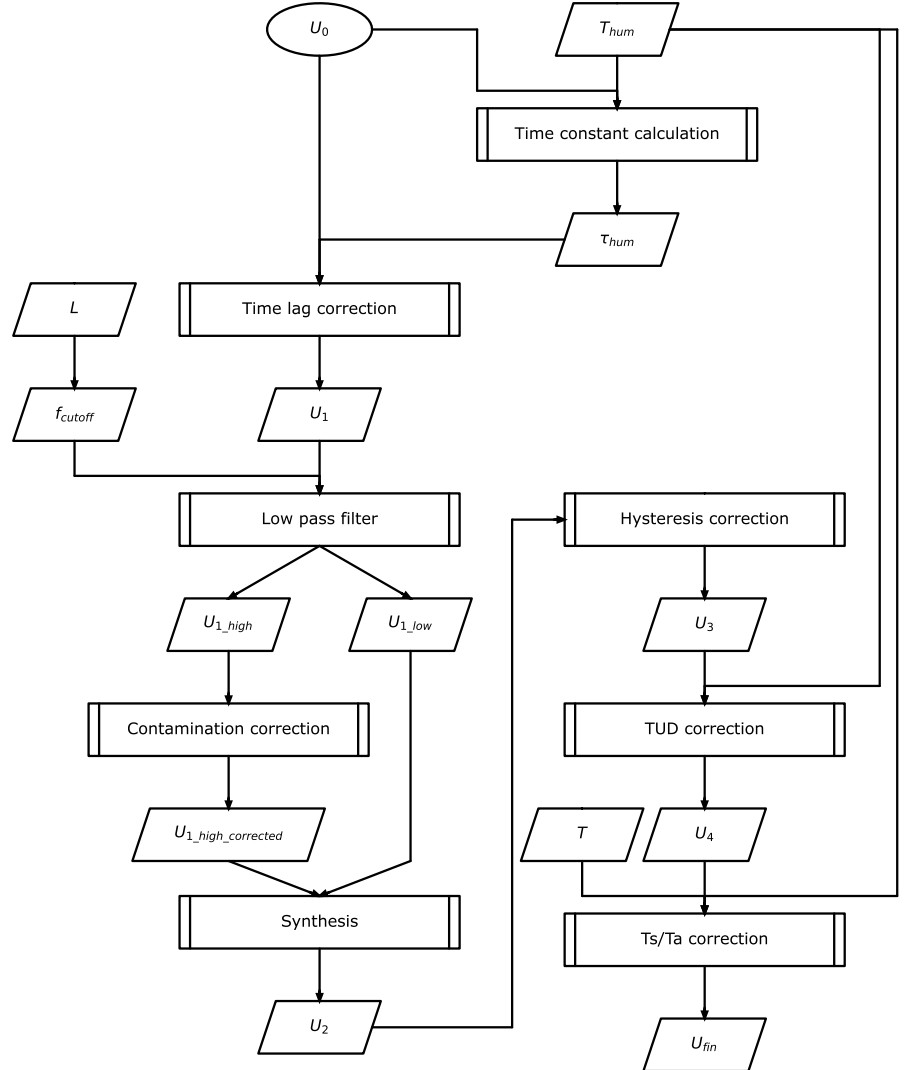

**Figure 5.** As per Fig.3, but for RH

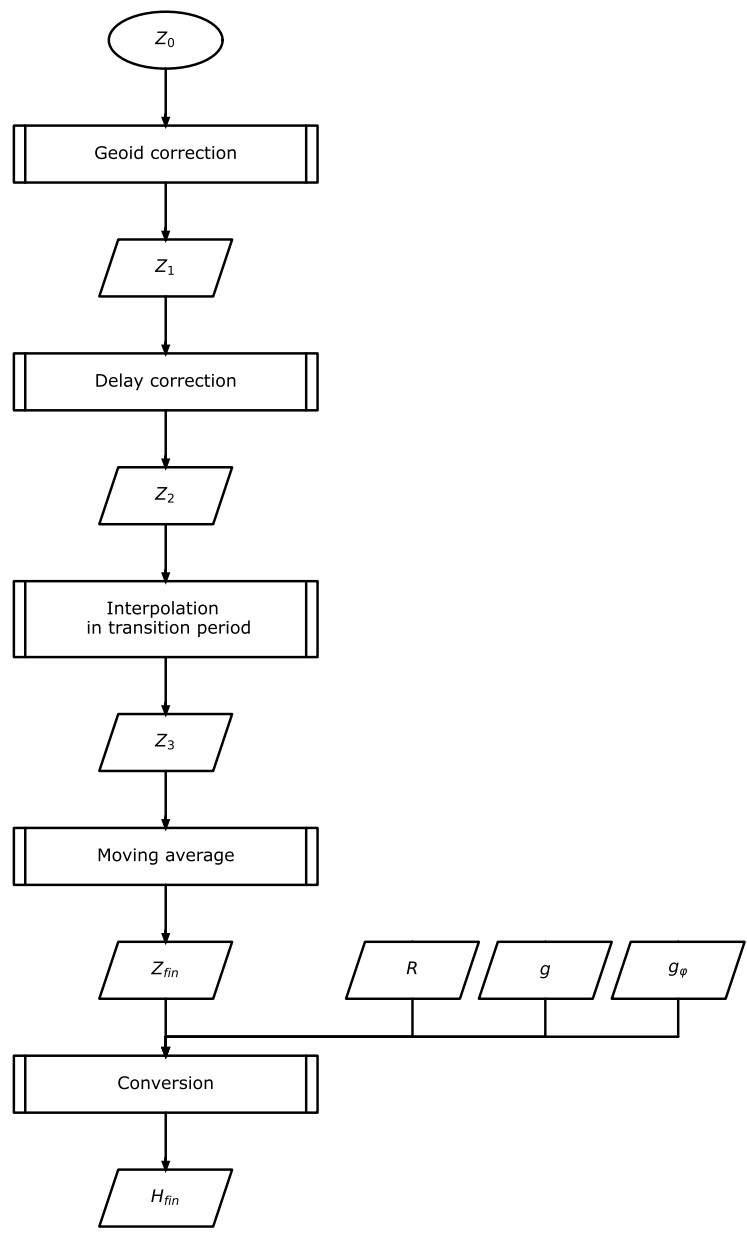

**Figure 6.** As per Fig.3, but for geometric altitude, $Z$, and geopotential height, $H$

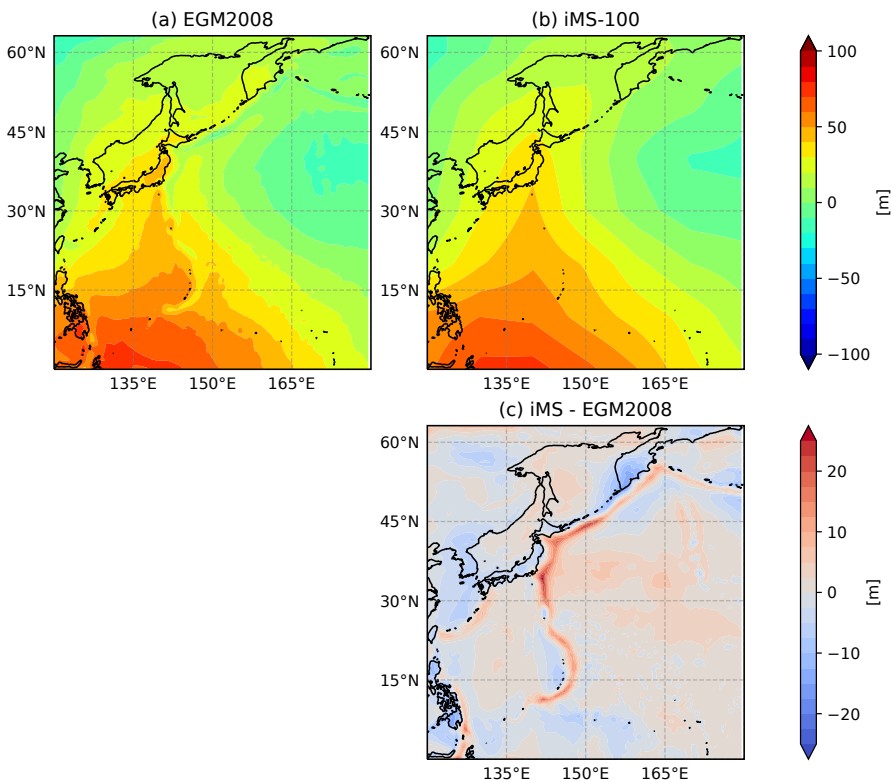

**Figure 7.** (a) Geoid height for IMS-100-GDP (EGM2008, 5 ′ × 5 ′), (b) IMS-100 original (10° × 10°), and (c) the difference (Northwest Pacific basin)

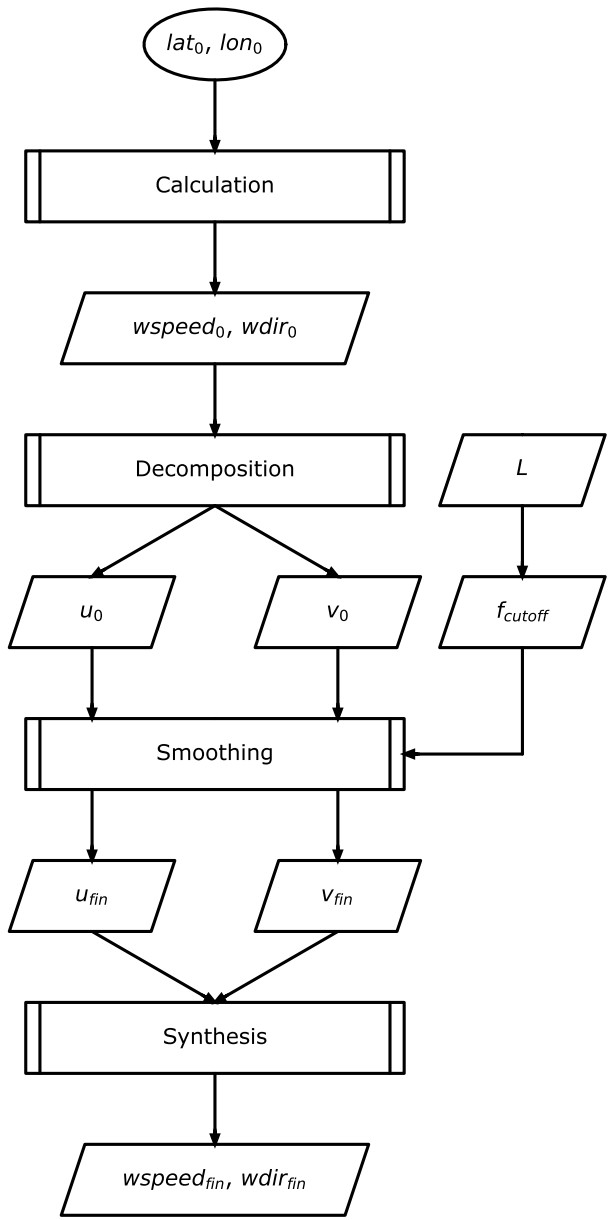

**Figure 8.** As per Fig.3, but for wind.

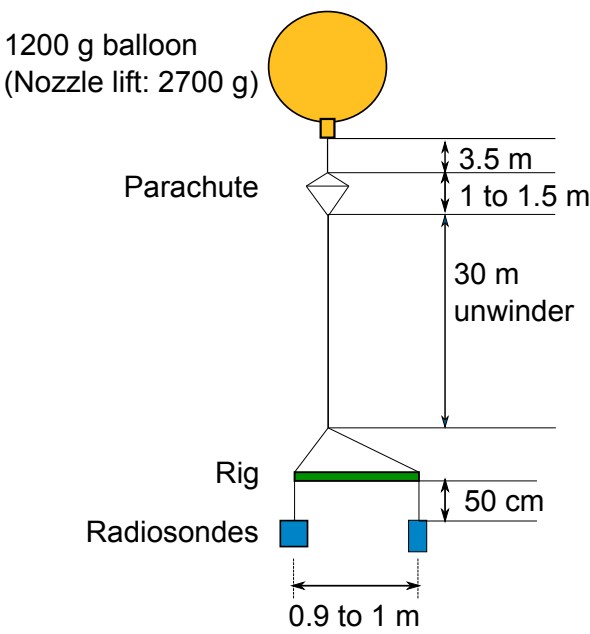

**Figure 9.** Payload configurations for dual sounding

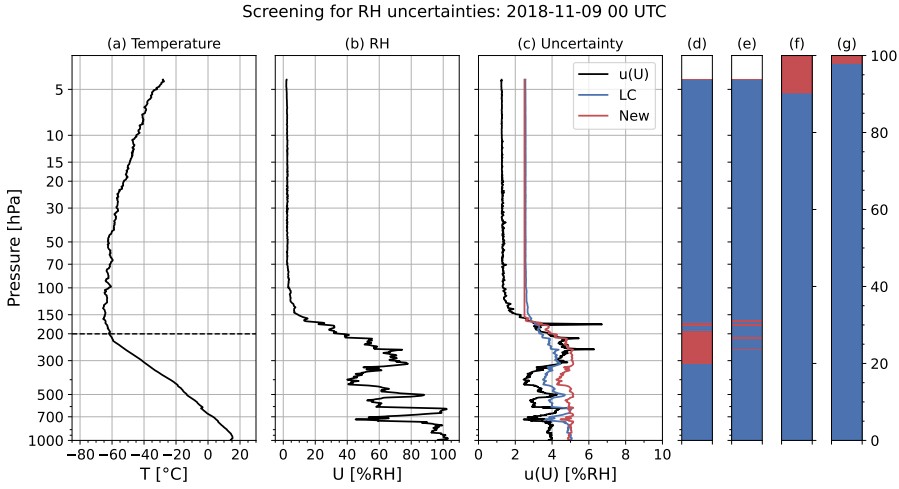

**Figure 10.** The profile of (a) temperature, (b) RH and (c) the uncertainties for RH at 00 UTC on 9 November, 2018. The horizontal dashed line in (a) represents the troropause. The blue and red lines in (c) show the criteria of screening according to Eqs.14 and 16. Panels (d) and (e) show whether the data within (blue) or exceeding (red) the criteria by Eqs.14 and 16 at each altitude, respectively. Panels (f) and (g) show the percentage of data within or exceeding in the whole profile, thus, equivalent to the rearrangement of panels (d) and (e).

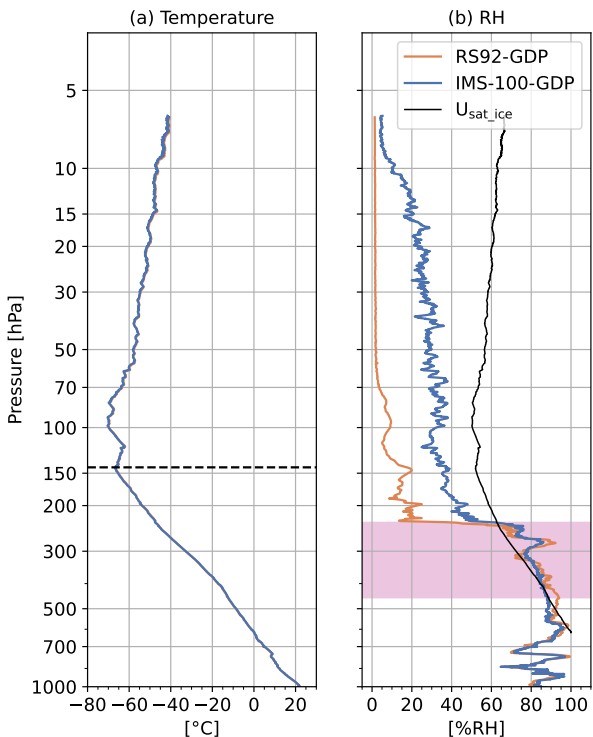

**Figure 11.** Temperature (a) and RH (b) profiles for RS92-GDP (orange) and IMS-100-GDP (blue) at 12UTC on 22 September 2017. The dashed line in (a) represents tropopause pressure. The black line in (b) shows the RH profile for ice saturated air with IMS-100-GDP temperature; air is supersaturated for ice if the observed RH exceeds this value. Pink shaded layer (453.2 – 232.4 hPa) shows the ISSR for IMS-100-GDP.

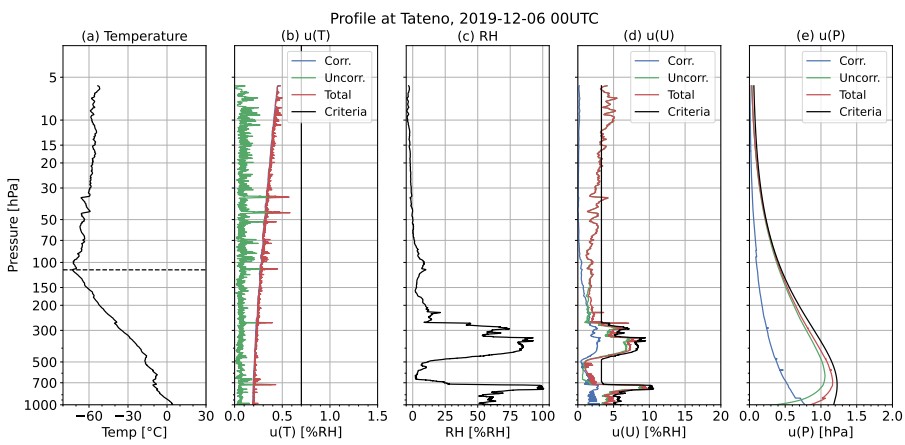

**Figure 12.** As per Fig.4, but for the case at 00 UTC on 6 December, 2019.

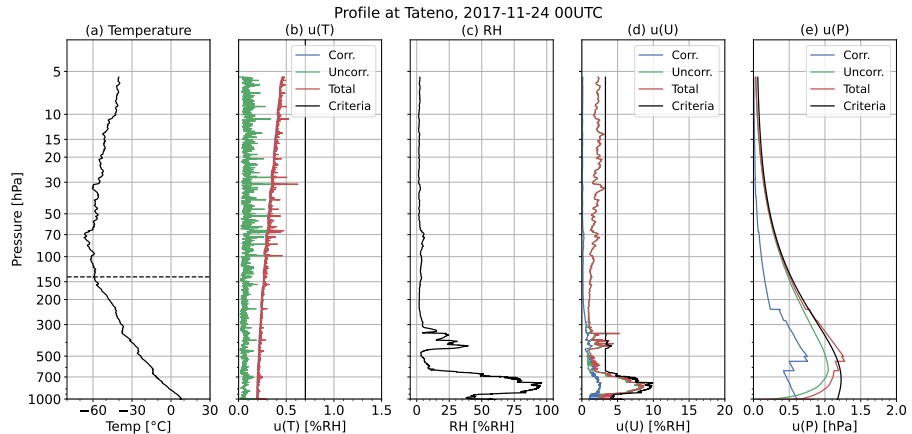

**Figure 13.** As per Fig.4, but for the case at 00 UTC on 24 November, 2017.

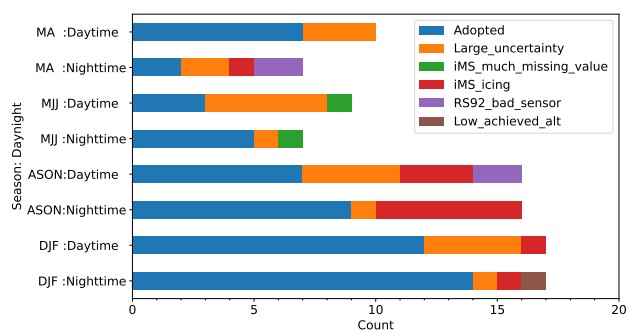

**Figure 14.** Major factors in data screening. "Adopted" is used for comparison

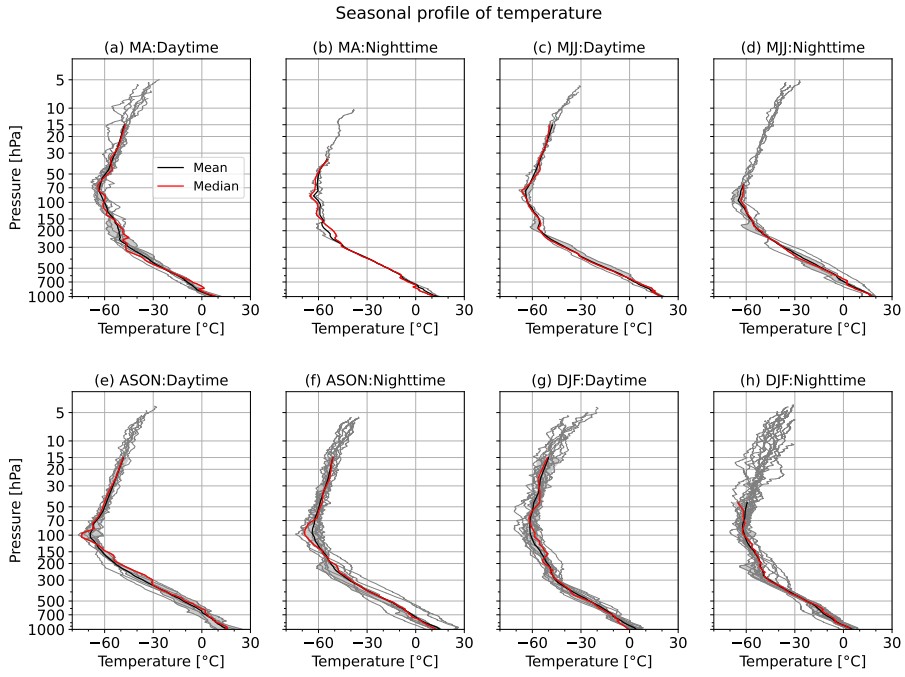

**Figure 15.** Seasonal profiles of IMS-100-GDP. Gray lines represent profiles for individual flights, red represents profile means, and black represents profile medians. Shading represents 25 – 75 percentiles.

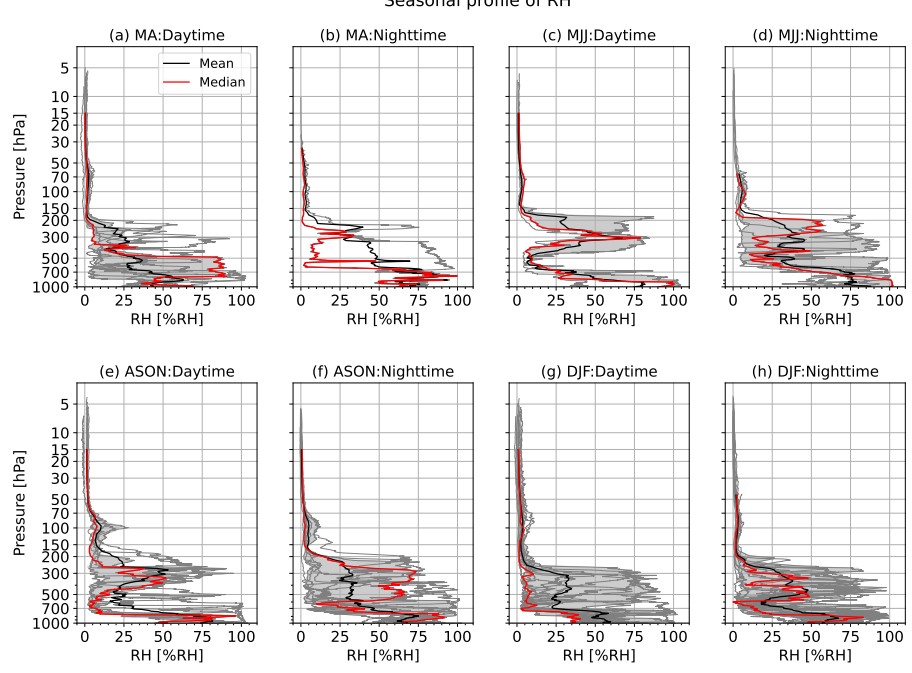

**Figure 16.** As per Fig. 15, but for RH

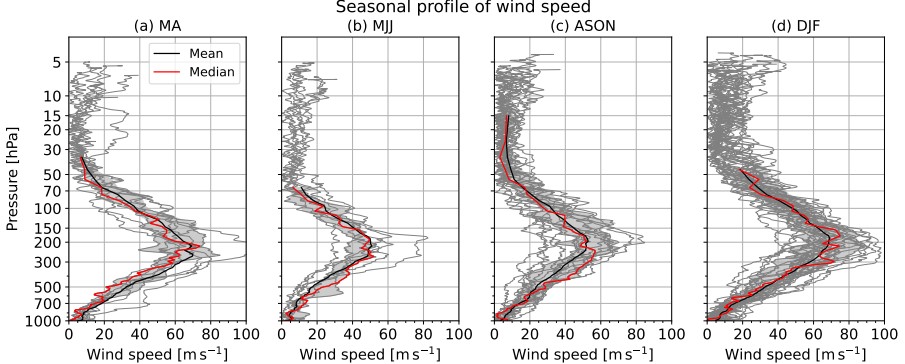

**Figure 17.** As per Fig. 15, but for wind speed

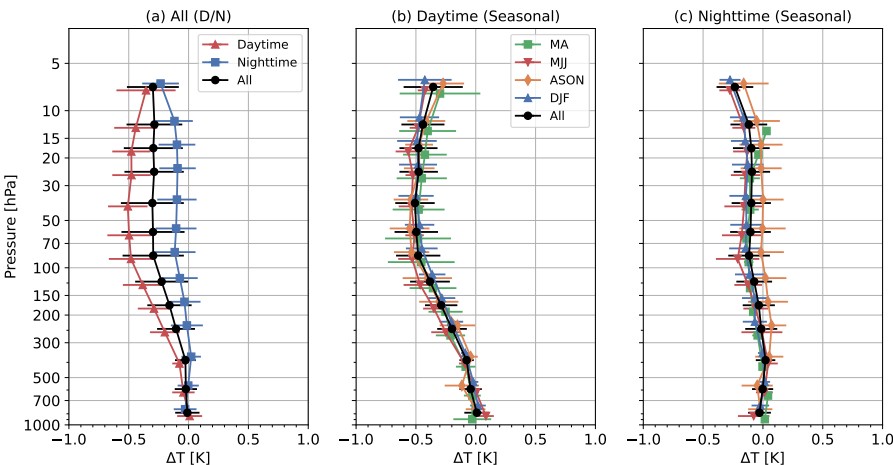

**Figure 18.** (a) Profiles of ensemble mean temperature differences (IMS-100-GDP minus RS92-GDP) with standard deviations (error bars) for all seasons combined. Red, blue and black lines represent daytime, nighttime and all data, respectively. Points are vertically shifted for ease of viewing. (b) As per (a), but representing seasonal profiles for daytime data. The green, red, orange, blue and black lines represent MA, MJJ, ASON, DJF and all seasons, respectively. (c) As per (b), but for nighttime data

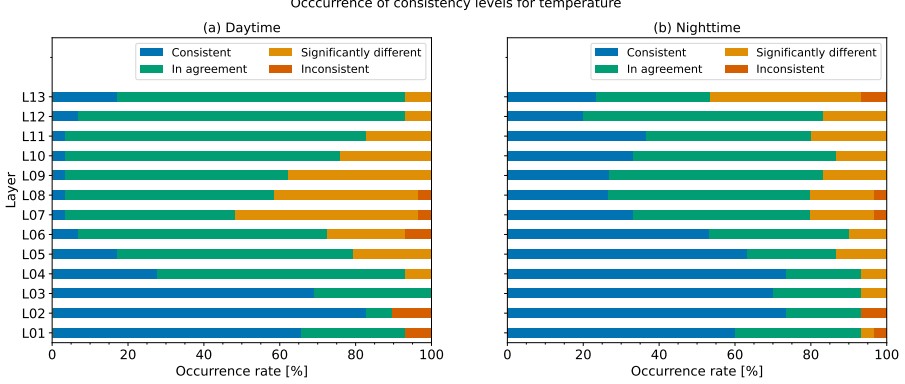

**Figure 19.** Temperature consistency rank for individual pressure layers

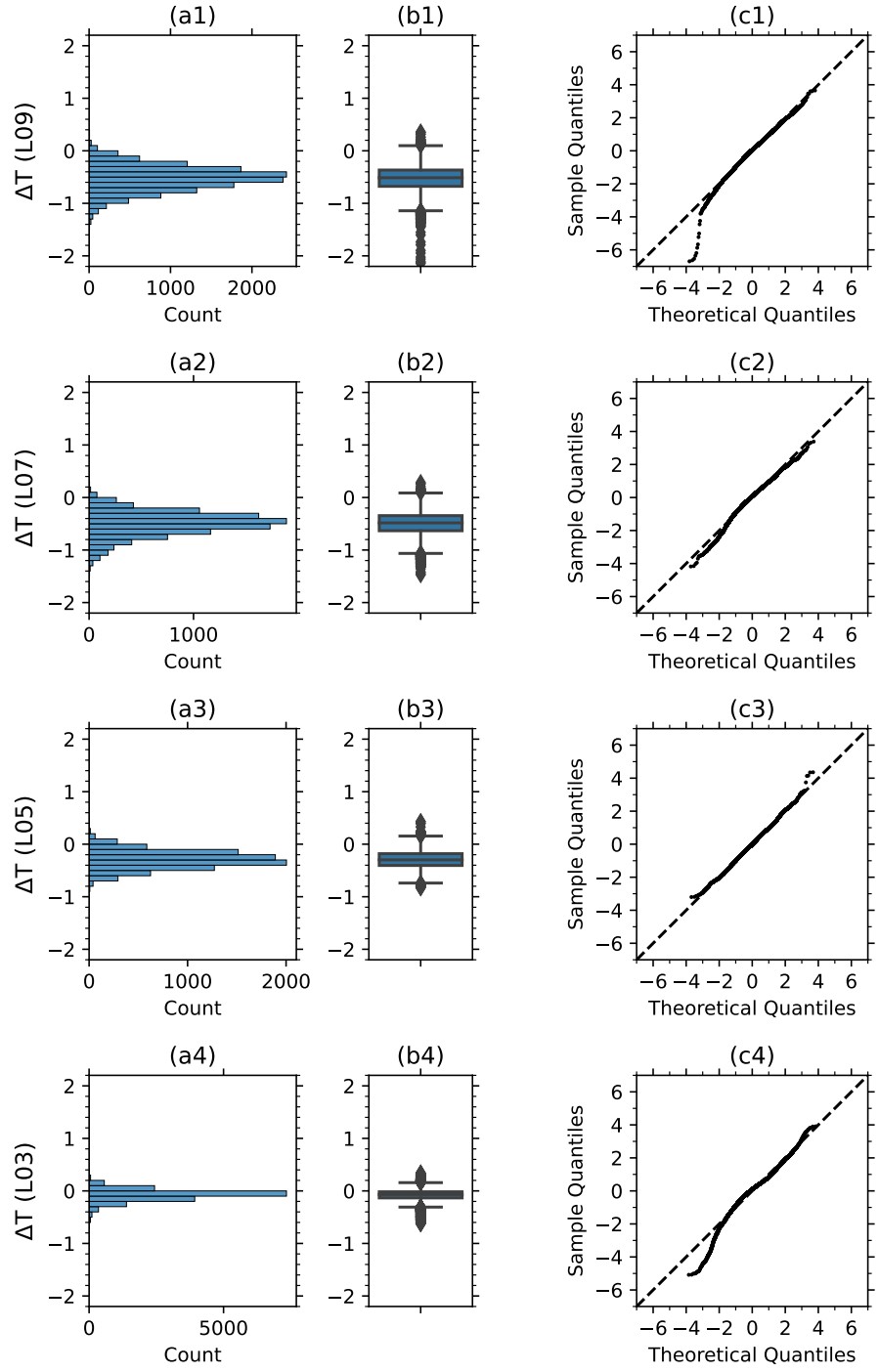

**Figure 20.** Distribution of temperature differences for daytime observations. aX: histogram; bX: boxplots; and cX: quantile-quantile plots. X indicates layer for (1) L09 (50 − 30 hPa), (2) L07 (100 − 70 hPa), (3) L05 (200 − 150 hPa), and (4) L03 (500 − 300 hPa)

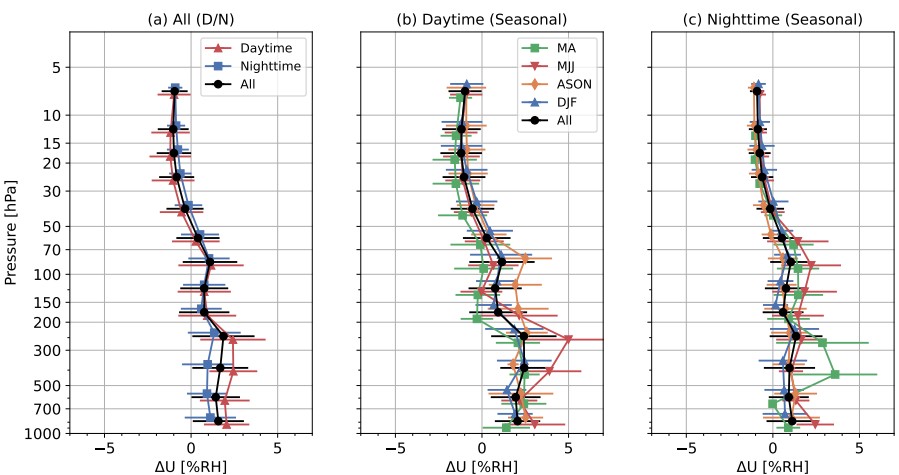

**Figure 21.** As per Fig. 18, but for RH

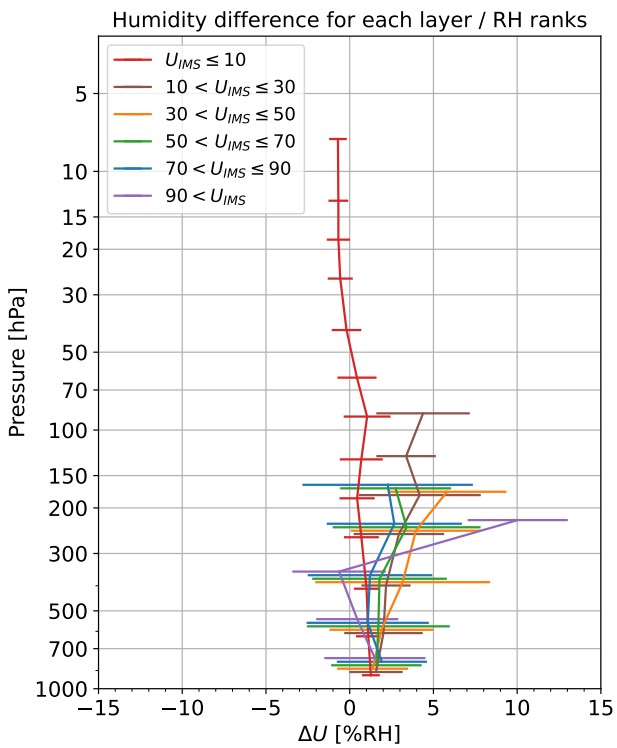

**Figure 22.** Mean RH difference for six RH ranks

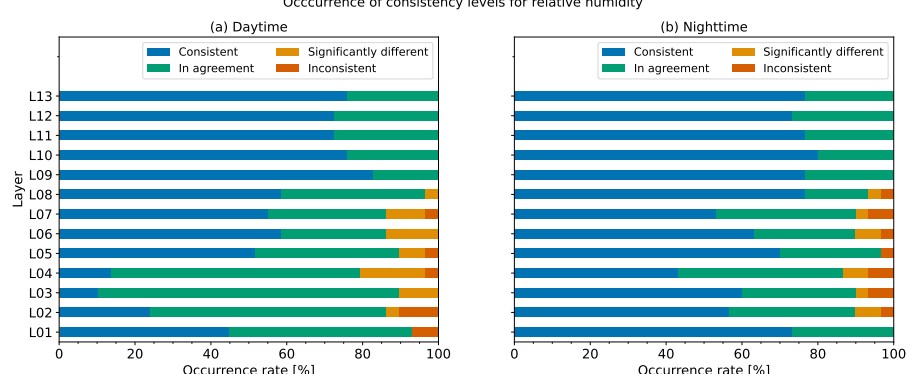

**Figure 23.** As per Fig.19, but for RH

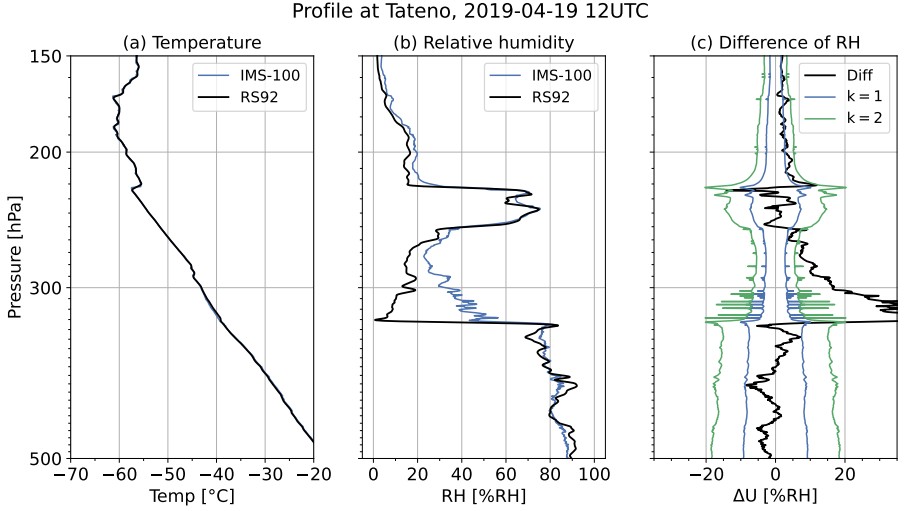

**Figure 24.** Comparison between 500 and 150 hPa levels (L3 to L5) at 12 UTC (21 LT) on 19 April 2019: (a) temperature, (b) RH, (c) RH difference

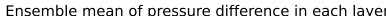

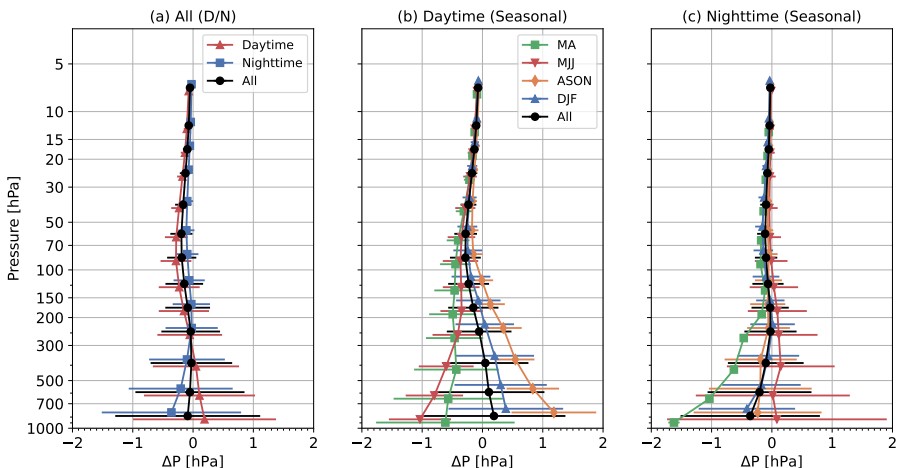

**Figure 25.** As per Fig. 18, but for pressure

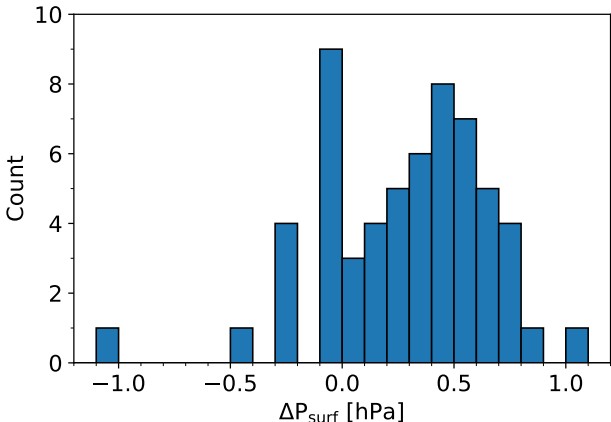

**Figure 26.** Histogram for the surface pressure difference (RS92 minus barometer values)

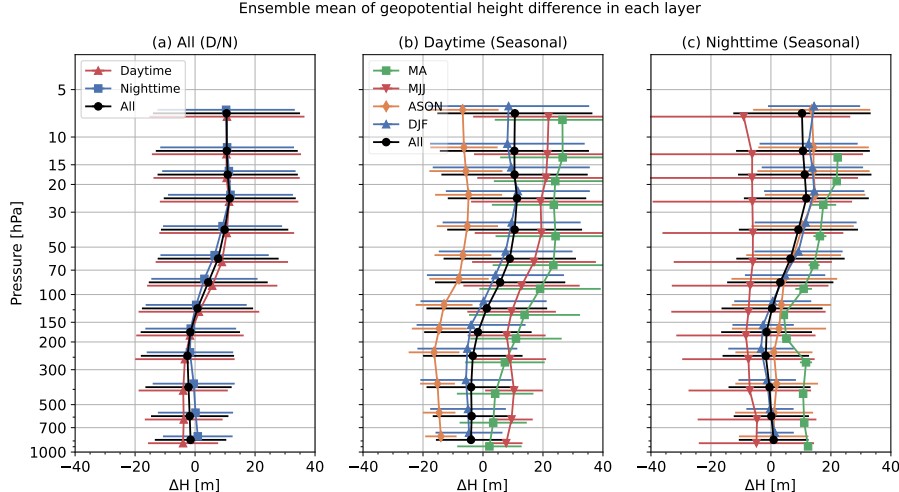

**Figure 27.** As per Fig. 18, but for geopotential height

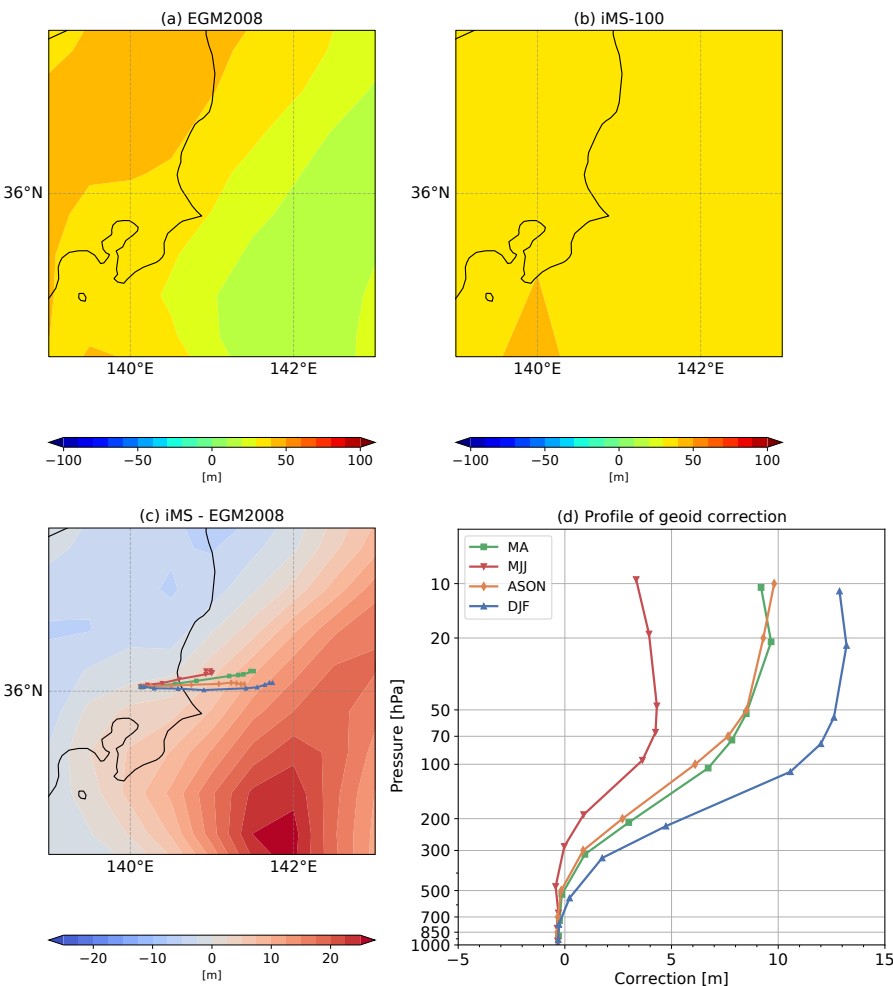

**Figure 28.** (a) Geoid height for IMS-100-GDP, (b) iMS-100 original, (c) difference of geoid height and seasonal typical tracks, (d) correction values along typical tracks

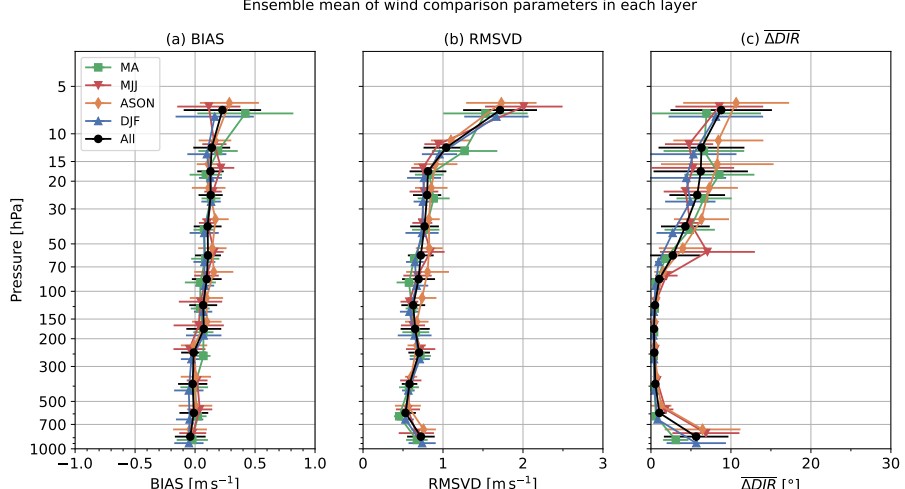

**Figure 29.** Seasonal wind difference parameters with coloring as per Fig. 18(b): (a) BIAS, (b) RMSVD (c) $\overline{\Delta DIR}$.

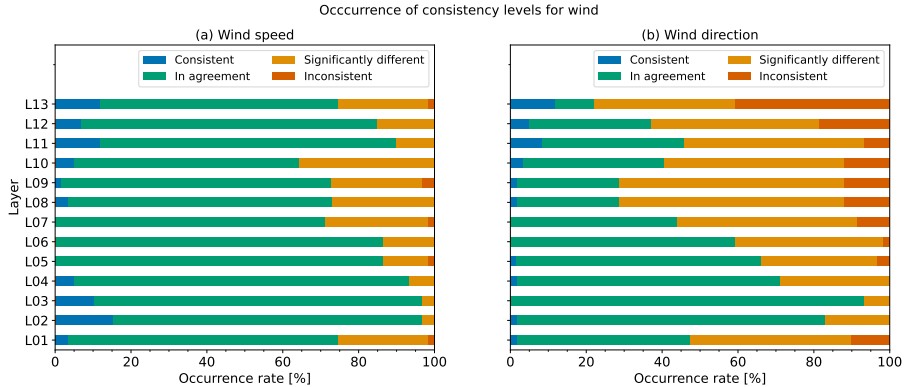

**Figure 30.** Wind consistency rank for individual pressure layers; (a) wind speed, (b) wind direction.

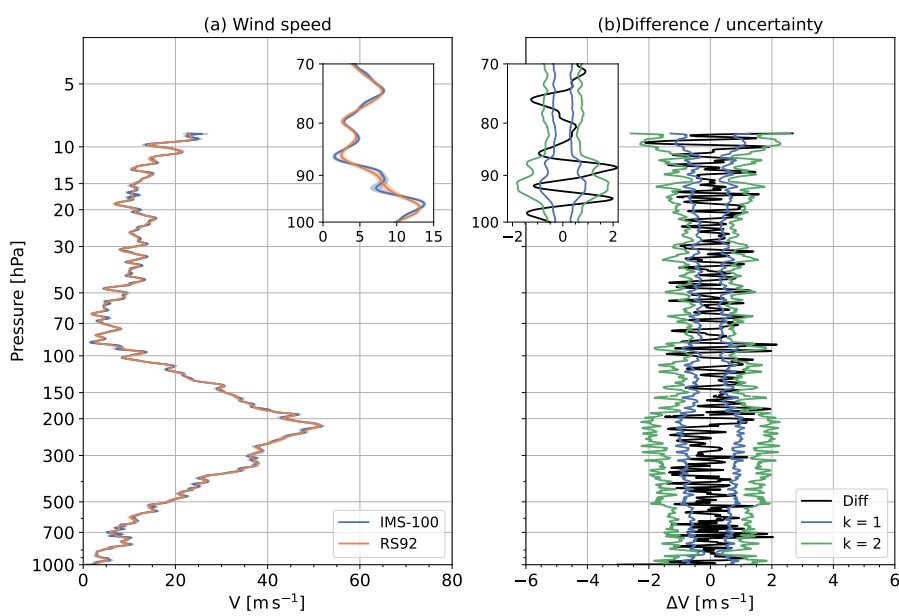

**Figure 31.** (a) Profiles of wind speed with uncertainty (shades) at 12 UTC on 12 July, 2019. Blue and orange lines represent IMS-100-GDP and RS92-GDP, respectively. (b) Profiles of wind speed difference (IMS-100-GDP minus RS92-GDP) (black line) and synthetic uncertainty (blue and green lines for $k = 1$ and $k = 2$, respectively)

.

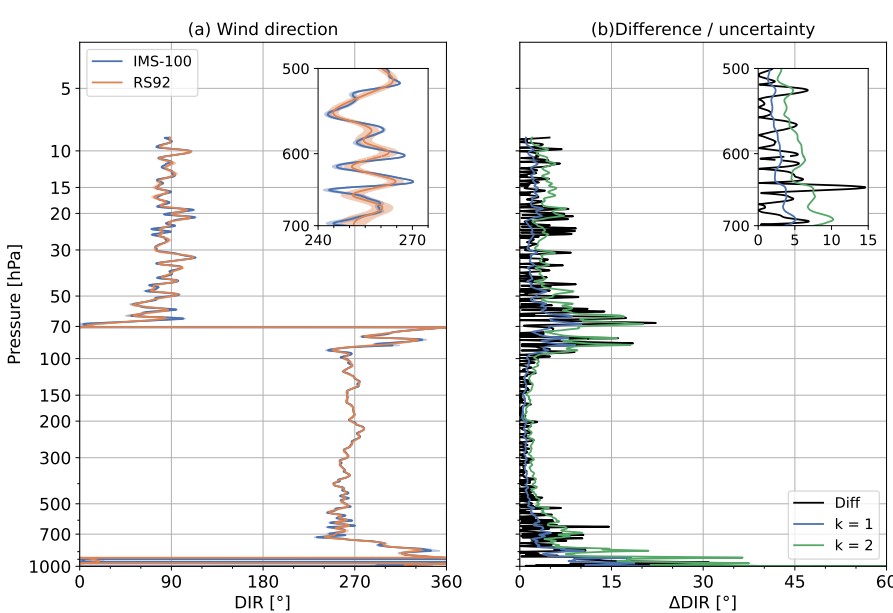

**Figure 32.** As per Fig. 31, but for wind direction. The differences in wind direction are calculated as the angle between the wind vectors for both GDPs, which is not considered clockwise / counterclockwise.

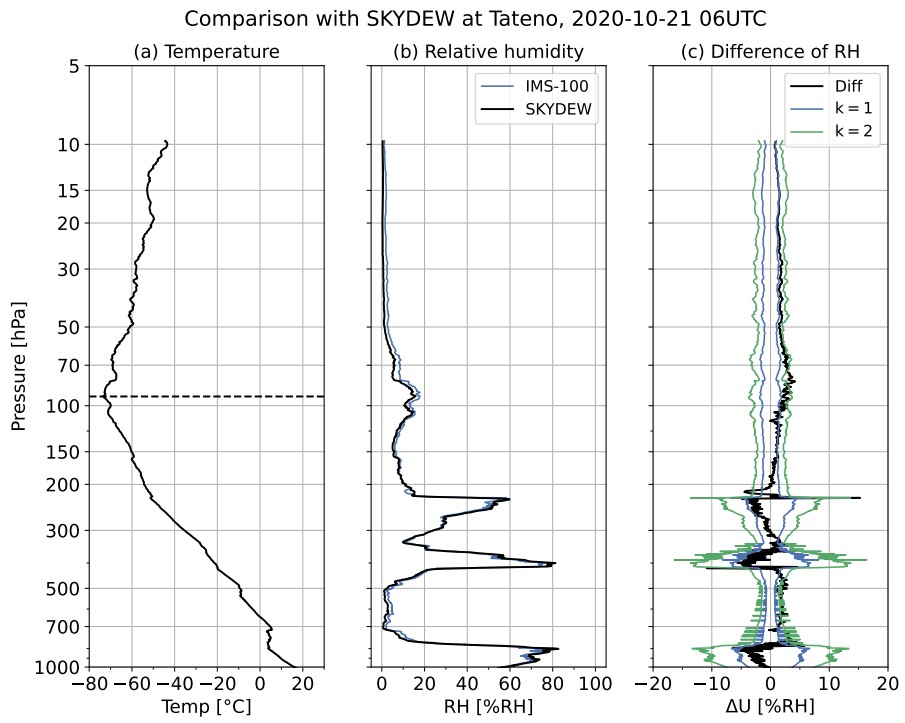

**Figure 33.** Comparison of RH profiles between IMS-100-GDP and SKYDEW at 06 UTC (15 LT) on 21 October 2020. (a) Air temperature from iMS-100. The dashed line shows the tropopause height (92.2 hPa; 16983.0 m in geopotential height). (b) RH from both. The blue and black lines show IMS-100-GDP and SKYDEW, respectively. SKYDEW RH is calculated from SKYDEW frost/dew-point temperature and iMS-100 air temperature. (c) RH difference in iMS-100 from SKYDEW (black) and RH uncertainty for IMS-100-GDP (blue/green for $k = 1$ and 2, respectively).

**Table 1.** Specifications of radiosondes (Vaisala Oyj., 2013; Meisei Electric Co., Ltd., 2020; Dirksen et al., 2014; Hoshino et al., 2022a)

| Radiosonde | iMS-100 | RS92 |
| --- | --- | --- |
| **Temperature** | | |
| Sensor type | Thermistor | Capacitive wire |
| Range | -90 °C to +60 °C | -90 °C to +60 °C |
| Resolution | 0.1 °C | 0.1 °C |
| **Humidity** | | |
| Sensor type | electrostatic capacitance humidity sensor | thin-film capacitor, headed twin sensor |
| Range | 0 %RH to 100 %RH | 0 %RH to 100 %RH |
| Resolution | 0.1 %RH | 1 %RH |
| Saturation vapor pressure formulation | Hyland and Wexler equation (Hyland and Wexler, 1983) | |
| **Pressure / geopotential height** | | |
| Sensor type | Calculated from GPS altitude | Silicon pressure sensor and calculated from GPS altitude |
| Pressure range | 1050 hPa to 3 hPa | 1080 hPa to 3 hPa |
| Pressure resolution | 0.1 hPa | 0.1 hPa |
| Calculation | Pressure is calculated from the GPS geopotential height using the hypsometric equation | In the lower part of the profile: the pressure sensor is used, and the geopotential height is derived from pressure using the hypsometric equation. In the upper part of the profile: use the GPS sensor |
| **Wind** | | |
| Sensor / calculation | Motion vector from GPS positioning (with SBAS)[1] | GPS wind finding (with GBAS) |
| Dimension (DWH) | 53 mm × 55 mm × 131 mm | 75 mm × 80 mm × 220 mm |
| Weight (with batteries) | 40 g [2] | 290 g |
| Ground system | MGPS2 (version 3 or higher) | DigiCORA III (version 3.6.4) MW41 |

[1] Wind also can be derived from Doppler shift of GPS signal, like RS-11G. [2] Weight for biodegradable materia model, while 38 g for EPS model.

**Table 2.** Sources contributing to iMS-100 temperature measurement uncertainty.

| Source | Value | (Un)correlated |
|---|---|---|
| Thermistor calibration, $u_{\text{Tcal1}}$ | $0.3/\sqrt{3}$ | correlated |
| Variation in temperature in calibration chamber, $u_{\text{Tcal2}}$ | $0.13/\sqrt{3}$ | correlated |
| Spike correction, $u_{\text{spike}}$ | $\sigma(\Delta T_{\text{spike}})$ | uncorrelated |
| Moving average, $u_{\text{ma}}$ | | uncorrelated |
| Albedo for radiation correction, $u_{\text{albedo}(T)}$ | $\left\|\Delta T_{\text{albedo}=90\%} - \Delta T_{\text{albedo}=10\%}\right\|/(2\sqrt{3})$ | correlated |
| Ventilation for radiation correction, $u_{\text{vent}(T)}$ | $\left\|\Delta T(\overline{\text{asc}}) - \Delta T(\text{vent}_{\text{pt}})\right\|/\sqrt{3}$ $\Delta T(\text{vent}_{\text{pt}}) = \sqrt{\text{asc}_{\text{raw}}^2 + \sigma_{\text{wind}}^2}$ | correlated |
| Sensor orientation, $u_{\text{orien}}(T)$ | | uncorrelated |
| Correlated uncertainty, $u_{\text{cor}}(T)$ | $\sqrt{u_{\text{Tcal1}}^2 + u_{\text{Tcal2}}^2 + u_{\text{albedo}(T)}^2 + u_{\text{vent}(T)}^2}$ | |
| Uncorrelated uncertainty, $u_{\text{ucor}}(T)$ | $\sqrt{u_{\text{spike}}^2 + u_{\text{ma}}^2 + u_{\text{orien}}^2}$ | |
| Total uncertainty, $u(T)$ | $\sqrt{u_{\text{cor}}^2(T) + u_{\text{ucor}}^2(T)}$ | |

**Table 3.** Coefficients for TUD correction (Eq.3).

| Symbol | Coefficients |
|---|---|
| $K_{0,0}$ | $2.155 \times 10^{-2}$ |
| $K_{0,1}$ | $4.961 \times 10^{-3}$ |
| $K_{0,2}$ | $-0.888 \times 10^{-5}$ |
| $K_{1,1}$ | $-67.70$ |
| $K_{1,2}$ | $1.624 \times 10^{-1}$ |
| $K_{2,0}$ | $30.89$ |
| $K_{2,1}$ | $1.798 \times 10^{-2}$ |
| $K_{3,0}$ | $-1.536$ |
| $K_{3,1}$ | $-2.074 \times 10^{-1}$ |
| $K_{3,2}$ | $8.713 \times 10^{-4}$ |

**Table 4.** Coefficients for saturated water vapor in Eq. 6

| a | Liquid | Ice |
|---|---|---|
| 0 | -0.58002206 $\times 10^4$ | -0.56745359 $\times 10^4$ |
| 1 | 0.13914993 $\times 10^1$ | 0.63925247 $\times 10^1$ |
| 2 | -0.48640239 $\times 10^{-1}$ | -0.96778430 $\times 10^{-2}$ |
| 3 | 0.41764768 $\times 10^{-4}$ | 0.62215701 $\times 10^{-6}$ |
| 4 | -0.14452093 $\times 10^{-7}$ | 0.20747825 $\times 10^{-8}$ |
| 5 | 0 | -0.94840240 $\times 10^{-12}$ |
| 6 | 0.65459673 $\times 10^1$ | 0.41635019 $\times 10^1$ |

**Table 5.** Sources contributing to iMS-100 relative humidity measurement uncertainty.

| Source | Value |
|---|---|
| Sensor calibration, $u_{\mathrm{Ucalib}}$ | $2/\sqrt{3}$ |
| Time-lag correction, $u_{\mathrm{TL}}(U)$ | $\dfrac{\left|U(\tau + u_\tau) - U(\tau - u_\tau)\right|}{2\sqrt{3}}$ $u_\tau = 0.25\tau$ |
| Low-pass filtering in contamination correction, $u_{\mathrm{LPF}}$ | |
| IIR filtering for high-pass component in contamination correction, $u_{\mathrm{IIR}}$ | |
| Hysteresis correction, $u_{\mathrm{hys}}(U)$ | $u_{\tau\_\mathrm{hys}} = \dfrac{\left|U(\tau_{\mathrm{hys}} + u_{\tau_{\mathrm{hys}}}) - U(\tau_{\mathrm{hys}} - u_{\tau_{\mathrm{hys}}})\right|}{2\sqrt{3}}$ $u_{\alpha\_\mathrm{hys}} = \dfrac{\left|U(\alpha_{\mathrm{hys}} + u_{\alpha_{\mathrm{hys}}}) - U(\alpha_{\mathrm{hys}} - u_{\alpha_{\mathrm{hys}}})\right|}{2\sqrt{3}}$ $u_{\mathrm{hys}}(U) = \sqrt{u_{\tau\_\mathrm{hys}}^2(U) + u_{\alpha\_\mathrm{hys}}^2(U)}$ |
| TUD correction, $u_{\mathrm{TUD}}(U)$ | Propagated from $u_{\mathrm{hys}}(U)$ |
| Ts/Ta correction, $u_{\mathrm{TsTa}}(U)$ | Propagated from $u_{\mathrm{TUD}}(U)$ |
| Correlated uncertainty, $u_{\mathrm{cor}}(U)$ | $\sqrt{u_{\mathrm{Ucalib}}^2 + u_{\mathrm{TsTa}}^2(U)}$ |
| Total uncertainty, $u(U)$ | $\sqrt{u_{\mathrm{cor}}^2(U) + u_{\mathrm{LPF}}^2(U) + u_{\mathrm{IIR}}^2(U)}$ |

 **Table 6.** List of dual sounding events with iMS-100 and RS92 used for comparison. Weather code is according to WMO code table 4677.

| # | Scheduled date | time (UTC) | Weather | Clouds | | | | Achieved level (iMS-100) | | Remarks |
|---|---|---|---|---|---|---|---|---|---|---|
| | | | | N | $C_L$ | $C_M$ | $C_H$ | Height [m] | Pressure [hPa] | |
| 1 | 15 September 2017 | 00:00 | 02 | 7 | 1 | 3 | 2 | 35406.7 | 5.4 | |
| 2 | 29 September 2017 | 00:00 | 02 | 1 | 1 | 3 | 0 | 35290.6 | 5.5 | |
| 3 | 10 November 2017 | 00:00 | 02 | 4 | 5 | 0 | 0 | 35950.1 | 4.8 | |
| 4 | 17 November 2017 | 12:00 | 02 | 7 | 1 | 3 | 2 | 33931.2 | 6.4 | |
| 5 | 1 December 2017 | 12:00 | 02 | 7 | 5 | / | / | 35672.9 | 4.9 | |
| 6 | 8 December 2017 | 00:00 | 03 | 7 | 0 | 3 | 8 | 34446.7 | 5.9 | |
| 7 | 15 December 2017 | 12:00 | 02 | 7 | 5 | / | / | 34188.4 | 6.1 | |
| 8 | 22 December 2017 | 00:00 | 10 | 6 | 5 | 0 | 2 | 33848.8 | 6.6 | |
| 9 | 29 December 2017 | 12:00 | 02 | 6 | 0 | 7 | 0 | 34395.5 | 6.1 | |
| 10 | 5 January 2018 | 00:00 | 02 | 7 | 5 | 3 | 2 | 35321.5 | 5.2 | |
| 11 | 19 January 2018 | 00:00 | 10 | 3 | 0 | 3 | 1 | 31732.5 | 8.8 | |
| 12 | 26 January 2018 | 12:00 | 02 | 0 | 0 | 0 | 0 | 35054.1 | 5.3 | |
| 13 | 2 February 2018 | 00:00 | 85 | 8 | 2 | / | / | 35134.6 | 5.4 | |
| 14 | 9 February 2018 | 12:00 | 03 | 7 | 0 | 3 | 2 | 35864.0 | 4.9 | |
| 15 | 23 February 2018 | 12:00 | 11 | 0 | 0 | 0 | 0 | 34358.1 | 5.6 | |
| 16 | 2 March 2018 | 00:00 | 02 | 1 | 0 | 0 | 1 | 33304.7 | 6.6 | |
| 17 | 30 March 2018 | 00:00 | 02 | 1 | 0 | 0 | 2 | 32382.3 | 8.1 | |
| 18 | 13 April 2018 | 00:00 | 02 | 0 | 0 | 0 | 0 | 34787.8 | 5.6 | |
| 19 | 4 May 2018 | 12:00 | 02 | 7 | 2 | / | / | 35430.0 | 5.4 | |
| 20 | 25 May 2018 | 00:00 | 02 | 7 | 2 | / | / | 34918.9 | 5.8 | |
| 21 | 1 June 2018 | 12:00 | 02 | 7 | 5 | / | / | 36030.3 | 5.1 | |
| 22 | 17 August 2018 | 00:00 | 02 | 4 | 1 | 0 | 1 | 35039.7 | 5.8 | |
| 23 | 28 September 2018 | 00:00 | 28 | 8 | 6 | / | / | 33271.6 | 7.3 | |
| 24 | 19 October 2018 | 12:00 | 25 | 7 | 2 | 7 | / | 34918.4 | 5.6 | |
| 25 | 2 November 2018 | 12:00 | 02 | 7 | 8 | / | / | 33581.8 | 6.8 | |
| 26 | 9 November 2018 | 00:00 | 21 | 8 | 7 | / | / | 36797.5 | 4.3 | |
| 27 | 16 November 2018 | 12:00 | 02 | 7 | 2 | 3 | 2 | 33724.6 | 6.4 | |
| 28 | 30 November 2018 | 12:00 | 02 | 7 | 8 | / | / | 33938.2 | 6.2 | |
| 29 | 14 December 2018 | 12:00 | 02 | 1 | 5 | 0 | 0 | 33210.2 | 7.0 | |
| 30 | 4 January 2019 | 00:00 | 02 | 0 | 0 | 0 | 0 | 29925.7 | 10.9 | |
| 31 | 11 January 2019 | 12:00 | 05 | 0 | 0 | 0 | 0 | 35922.7 | 4.3 | |

| # | Scheduled date | Time (UTC) | Weather | Clouds | | | | Achieved level (iMS-100) | | Remarks |
|---|---|---|---|---|---|---|---|---|---|---|
| | | | | N | $C_L$ | $C_M$ | $C_H$ | Height [m] | Pressure [hPa] | |
| 32 | 18 January 2019 | 00:00 | 02 | 0 | 0 | 0 | 0 | 33459.6 | 6.4 | |
| 33 | 25 January 2019 | 12:00 | 03 | 7 | 0 | 7 | / | 20897.7 | 44.6 | |
| 34 | 1 February 2019 | 00:00 | 02 | 0 | 0 | 0 | 0 | 35588.1 | 5.0 | |
| 35 | 8 February 2019 | 12:00 | 02 | 2 | 2 | 0 | 0 | 34432.3 | 6.0 | |
| 36 | 22 February 2019 | 12:00 | 61 | 7 | 5 | 7 | / | 34368.1 | 6.0 | |
| 37 | 1 March 2019 | 00:00 | 80 | 8 | 2 | / | / | 32600.4 | 7.9 | |
| 38 | 15 March 2019 | 00:00 | 02 | 1 | 2 | 0 | 0 | 35778.2 | 5.0 | |
| 39 | 22 March 2019 | 12:00 | 02 | 7 | 2 | 3 | / | 30755.7 | 10.3 | [1] |
| 40 | 29 March 2019 | 00:00 | 02 | 2 | 5 | 7 | / | 35420.7 | 5.4 | |
| 41 | 12 April 2019 | 00:00 | 02 | 7 | 5 | 7 | 2 | 34351.5 | 6.2 | |
| 42 | 19 April 2019 | 12:00 | 17 | 8 | 9 | / | / | 24879.8 | 25.4 | |
| 43 | 3 May 2019 | 12:00 | 02 | 7 | 0 | 3 | 2 | 19487.7 | 60.2 | |
| 44 | 10 May 2019 | 00:00 | 02 | 1 | 0 | 0 | 2 | 33723.9 | 6.8 | |
| 45 | 31 May 2019 | 12:00 | 02 | 7 | 0 | 7 | / | 35035.1 | 5.8 | |
| 46 | 21 June 2019 | 00:00 | 10 | 8 | 2 | 7 | / | 33677.6 | 7.1 | |
| 47 | 12 July 2019 | 12:00 | 21 | 8 | 0 | 7 | / | 32444.3 | 8.6 | |
| 48 | 23 August 2019 | 12:00 | 10 | 7 | 2 | 3 | / | 35233.8 | 5.8 | |
| 49 | 4 October 2019 | 12:00 | 02 | 1 | 0 | 3 | 0 | 32421.9 | 8.3 | |
| 50 | 15 November 2019 | 12:00 | 02 | 0 | 0 | 0 | 0 | 31786.2 | 8.8 | |
| 51 | 22 November 2019 | 00:00 | 25 | 8 | 8 | / | / | 33575.7 | 6.7 | |
| 52 | 29 November 2019 | 12:00 | 02 | 0 | 0 | 0 | 0 | 34727.1 | 5.6 | |
| 53 | 13 December 2019 | 12:00 | 02 | 7 | 5 | 0 | 2 | 30668.3 | 10.7 | |
| 54 | 20 December 2019 | 00:00 | 02 | 0 | 0 | 0 | 0 | 33485.9 | 6.6 | |
| 55 | 3 January 2020 | 00:00 | 02 | 0 | 0 | 0 | 0 | 33476.7 | 6.8 | |
| 56 | 10 January 2020 | 12:00 | 02 | 4 | 0 | 7 | 2 | 36964.1 | 4.2 | |
| 57 | 17 January 2020 | 00:00 | 02 | 8 | 8 | / | / | 31006.0 | 9.8 | |
| 58 | 24 January 2020 | 12:00 | 10 | 1 | 0 | 3 | 2 | 35957.1 | 4.7 | |
| 59 | 31 January 2020 | 00:00 | 02 | 1 | 5 | 0 | 0 | 36588.2 | 4.4 | |

[1] Truncated due to excessive temperature gaps. RS92 achieved 34434.6 m (6.1 hPa).

**Table 7.** Pressure range for allocation of iMS-100 - RS92 differences based on RS92 pressure (i.e., $\mathrm{bottom} \geq P_i^{\mathrm{RS92}} > \mathrm{top}$)

| Layer # | 1 | 2 | 3 | 4 | 5 | 6 | 7 | 8 | 9 | 10 | 11 | 12 | 13 |
|---|---|---|---|---|---|---|---|---|---|---|---|---|---|
| Top [hPa] | 700 | 500 | 300 | 200 | 150 | 100 | 70 | 50 | 30 | 20 | 15 | 10 | 5 |
| Bottom [hPa] | 1000 | 700 | 500 | 300 | 200 | 150 | 100 | 70 | 50 | 30 | 20 | 15 | 10 |

**Table 8.** Terminology for for pair checking in independent measurements with identical quantities for consistency.

| $\lvert m_1 - m_2 \rvert < k\sqrt{u_1^2 + u_2^2}$ | TRUE | FALSE | significance level |
|---|---|---|---|
| $k = 1$ | consistent | suspicious | 32 % |
| $k = 2$ | in agreement | significantly different | 4.5 % |
| $k = 3$ | – | inconsistent | 0.27 % |