# Peer review of "Comparison of GRUAN Data Products for Meisei iMS-100 and Vaisala RS92 Radiosondes at Tateno, Japan"

_Atmospheric Measurement Techniques, 2021_

## Referee Comment (RC1)

Report on AMT-2021-374

This paper compares GDPs of 57 dual sounding flights related to Meisei iMS-100 and Vaisala RS92 radiosonde. T, RH, WS and WD are considered.

Overall, it is clearly written and has the potential to fill a gap in the understanding of iMS-100 measurement quality. The novelty is essentially in the data as the methods are quite standard. I think it is worth to be published after the comments below have been addressed.

Main comments
1) Overall, I find a logical/expositional gap between method section (§4) and result section (§5). In particular, §4 refers to layer averages and their standard deviation in the ensamble of M=57 dual flights (see below for more details). Instead in §5, following Immler's approach which the reader expects a consistency check at the single measurement level. So, these two levels should be better connected.
   It is worth noting that with known uncertainties called type B evaluation. For type A evaluation, Immler et al. (2010) suggests using Students't
2) I would like to see some more details about the Immler's consistency check mentioned in §4.5 and in Figure 15 (and 19).
3) What about missing values? (only large gaps mentioned in the introduction). Recently GRUAN community is considering the importance of interpolation and its uncertainty
   (see e.g. Fassò et al. (2020, *https://doi.org/10.5194/amt-13-6445-2020*) for T, and Colombo et al. (2022, https://iopscience.iop.org/article/10.1088/1361-6501/ac5bff/pdf) for RH ). These considerations should be present in the state of the art literature of the present manuscript.
4) The title focus on GDP comparison, an essential part of the GDP is the measurement uncertainty assessment. How do the uncertainties of the two products compare? I expected to see something about this

Typos and minor points
L.18: heare is here
L.37-38: a verb is missing?
L.40: RS-11G GDP is RS-11G GDP.1 ?
L.38-43: are these results from Kobayashi et al.? be more explicit
L.49: Sect.5 and Sect.6 add a comma
L.50: "See Appendix A for a summary …" the style is inconsistent with the rest of the paragraph, use passive form.
L.55: ant is and

L.245: "M in" is "M is"
L.245 Is M=57?, please be more explicit
L.252-253: To define a standard deviation, in Eq. (18) summands must be squared. After this, eq. (18) defines the standard deviation of the (mean) differences given by eq.(16). The ensamble mean difference is given by (17) and its standard deviation (taking the ensamble as a random sample) is not given by eq. 18, as stated in L.252 ("The standard deviation of the ensamble mean difference for individual pressure layers") but is given by sigma(A_k)/sqrt(M). Although the phrasing in L.252 is present in literature, it is imprecise and should read "the ensamble standard deviation of the mean differences"
L.255: "… error the …" is "… error of the …"

§4.5: this sect. is a 3-line section. For self-contentedness, I suggest to briefly report the Immler check, or to avoid heading this sentence as section.

---

## Referee Comment (RC2)

**Review of "Comparison of GRUAN Data Products for Meisei iMS-100 and Vaisala RS92 Radiosondes at Tateno, Japan" by Shunsuke Hoshino et al.**

The study undertakes a comparison of two radiosondes at the GRUAN site Tateno according to GRUAN principles of change management. Both sonde types have quantified uncertainties using the metrological principles of measurement traceability. The study is valuable both to GRUAN but also to the broader radiosonde user community in that it highlights issues around both sonde types. The journal is an appropriate target audience for this work and the study is clearly germane. The work in my view is publishable following corrections.

**Major comments**

1. The methods described in sections 2 through 4 are possibly in several cases a bit too brief and too heavily reliant upon the reader going back to and reading a number of previously published papers. Perhaps some key additional aspects need to be documented to ensure broad-scale methodological reproducibility. In particular section 4.5 is too brief given its overall importance in a GRUAN product comparison and should be expanded.
2. For figure 3 and subsequent similar figures it would be useful to describe in the figure 3 caption what the different shapes of the boxes denote? What is a circle, a parallelogram, a diamond and an oblong?
3. It would probably be worth spending some time discussing the very marked seasonality of the rejection rates shown in Figure 10 which is cited but not really discussed. Rejection rates are low in winter but very high in spring and summer. Why is this?
4. I always hate making this comment because it is immensely impressive for non-native speakers to produce papers in English. However, the paper would be much more readable if you could get a native speaker to edit for clarity. There are numerous places where minor edits would improve the readability and make the messaging stronger.

**Minor comments**

1. A reader would reasonably ask in the abstract why you mention 99 dual soundings but proceed to analyse only 57 of these. Can a few words be added to clarify why this 57 subset of the sample were analysed? Something like "Following data quality checks 57 flights were considered of sufficient data quality to produce GDP profiles and this subset is analysed"
2. Line 8 I would delete "with RS92-GDP" as this is already clear from earlier in the sentence
3. Line 19-20 I would write. "While the RS92-SGP radiosonde has a GDP it was required to seek alternative radiosonde models to use for operational reasons as the payloads often fall within the greater Tokyo metropolitan region and for health and safety reasons use of lighter instrumentation is necessary."

4. In lines 21-22 I would suggest making explicitly clear that the RS-11G has already been developed and certified as a GDP

5. Line 56 and not ant (typo)

6. Should line 105 not be 10' x 10' ? A geoid model at 10 degrees by 10 degrees feels implausibly coarse and the change to 5' by 5' later in paragraph is then huge.

7. Perhaps in 2.2.4 make clear why no pressure sensor is fitted. Presumably this is to save weight and because the errors in the hydrostatic equation based approximation are considered sufficiently small to justify the omission of such a sensor?

8. Lines 205-206 are not possible to follow logically. How did a single payload last 7 months? What is a logistic regression? Work required here to clarify this sentence please.

9. Lines 218-219 are again unclear. I think you mean something like: The criteria applied here apply solely to the present study and are not applied to the GDP. However, you also probably need to explain why this is the case. Surely the same processing should be applied to the GDP screening or is there some reason why this would not be possible?

10. The paragraph starting line 302 is presumably applicable to all meteorological elements measured and not just temperature. As such its placement here rather than elsewhere – perhaps most logically the discussion section of the paper – feels strange to me. There is already similar text on lines 398-405 so maybe you can just delete this?

11. Line 388-389 it is unclear for what parameter this finding applies. I assume pressure but it needs to be stated explicitly.

---

## Author Response (AR1)

Reply to RC1

Thank you very much for your reviewing our manuscript and providing us with valuable comments and suggestions. We reprocessed the IMS-100-GDP because we found some minor issues in the processing data used in the previous manuscript. Furthermore, we applied the updated screening of data following review comments. The number of samples has increased from 57 to 59. However, the main message of the manuscript has unchanged.

Hereafter, Cx represents the referee's comments and Rx represents the reply to Cx.

Main comments

C1) Overall, I find a logical/expositional gap between method section (§4) and result section (§5). In particular, §4 refers to layer averages and their standard deviation in the ensamble of M=57 dual flights (see below for more details). Instead in §5, following Immler's approach which the reader expects a consistency check at the single measurement level. So, these two levels should be better connected. It is worth noting that with known uncertainties called type B evaluation. For type A evaluation, Immler et al. (2010) suggests using Students't

R1) We have taken two approaches in comparing the two data products. The first method, the layer mean in the ensemble of dual flights, is to obtain a profile of differences between the products, which is necessary to create a homogenized data set for the climatological discussion at a site where there were instrument changes. The second method, the consistency verification using the uncertainty per single measurement level following Immler's approach (Type B evaluation), is necessary to know if the results obtained in the first verification can be regarded as being significantly different.

Also Immler et al. (2010) concluded that Type A evaluation of uncertainty is not expected to play an important role within GRUAN. Thus we did not discuss it in this study.

The explanation for concepts of the two methods is added to the beginning of Sect. 4 as follows:

"Temporally simultaneous measurements were compared using the two statistical approaches adopted by Kobayashi et al. (2019) to evaluate differences in the data products. The first method, the layer mean in the ensemble of dual flights (described in Section 4.4), is to obtain a profile of differences between products, which is necessary to create a homogenized data set for the climatological discussion at a site where there were instrument changes. The second method, the consistency verification using the uncertainty per single measurement level following type B evaluation in Immler et al. (2010) (described in Section 4.5), is necessary to know if the results obtained in the first method can be regarded as being significantly different."

Furthurmore, a brief note about Type A/B evaluation is added to Section 4.5 as follows:

"Immler et al. (2010) proposed an expression for the degree of consistency as shown in Table 8. This approach is a Type B evaluation of uncertainty. For Type A evaluation, Immler et al. (2010) concluded that it is not expected to play an important role within GRUAN, so it is not considered in this study."

C2) I would like to see some more details about the Immler's consistency check mentioned in §4.5 and in Figure 15 (and 19).

R2) The description of the consistency check is added as:

"Immler et al. (2010) proposed terminology for comparing pairs of independent measurements of the same quantity for consistency using estimated uncertainties as described in the following: Consider two independent measurements, m1 and m2, of the same measurand with standard uncertainties, u1 and u2, respectively. Assuming the hypothesis that m1 = m2 is true and uncertainty is normally distributed, the probability that occurs only by chance, is roughly 4.5% for k = 2 and 0.27% for k =3. If Eq. 32 is true for k = 2, it is very likely that the two measurements did in fact not measure the same thing, probably due to an unrecognized or unaccounted-for systematic effect in one or both measurements. Immler et al. (2010) proposed an expression for the degree of consistency as shown in Table 8. "

"For statistical consistency check, the total consistency ranks shown in Table 8 (1: consistent, 2: in agreement, 3: significantly different, or 4: inconsistent) between RS92 and iMS-100 within a specific pressure layer for a particular parameter are estimated as the 95 % percentile value of consistency ranking numbers within the layer."

C3) What about missing values? (only large gaps mentioned in the introduction). Recently GRUAN community is considering the importance of interpolation and its uncertainty (see e.g. Fassò et al. (2020, *https://doi.org/10.5194/amt-13-6445-2020*) for T, and Colombo et al. (2022, https://iopscience.iop.org/article/10.1088/1361-6501/ac5bff/pdf) for RH ). These considerations should be present in the state of the art literature of the present manuscript.

R3) Thank you for the references (we were aware of the work by Fassò et al. (2020), These are very good works; we are afraid, however, that the actual implementation to the GDPs will take more some time. In our current paper, we would like to limit the statement to a reference for future product development. The citation is added to the summary section as:

"The interpolation and the estimation of uncertainty for data missing periods are discussed in some articles. For example, Fassò et al. (2020) proposed a method for temperature data using the Gaussian process, and Colombo and Fassò (2022) attempted to apply it to RH data. These studies will be considered for future improved versions of the IMS-100-GDP. "

C4)   The title focus on GDP comparison, an essential part of the GDP is the measurement uncertainty assessment. How do the uncertainties of the two products compare? I expected to see something about this

R4) The uncertainty components and their estimation methods (some components depend on correction methods) vary by product and are not unified, so they are not simply comparable. Therefore, we have not discussed the comparison of uncertainties themselves.

Typos and minor comments

C5) L.18: heare is here

R5) Corrected.

C6) L.37-38: a verb is missing?

R6) Corrected.

C7) L.40: RS-11G GDP is RS-11G GDP.1?

R7) Corrected.

C8) L.38-43: are these results from Kobayashi et al.? be more explicit

R8) The reference is shown again in the revised manuscript.

C9) L.49: Sect.5 and Sect.6 add a comma

R9) Corrected.

C10) L.50: "See Appendix A for a summary ..." the style is inconsistent with the rest of the paragraph, use passive form.

R10) Corrected.

C11) L.55: ant is and

R11) Corrected.

C12) L.245: "M in" is "M is"

R12) Corrected.

C13) L.245 Is M=57?, please be more explicit

R13) Yes, but M is 59 due to reconsideration of the screening process. Revised as "M is the number of data sets, here 59."

C14) L.252-253: To define a standard deviation, in Eq. (18) summands must be squared. After this, eq. (18) defines the standard deviation of the (mean) differences given by eq.(16). The ensamble mean difference is given by (17) and its standard deviation (taking the ensamble as a random sample) is not given by eq. 18, as stated in L.252 ("The standard deviation of the ensemble mean difference for individual pressure layers") but is given by sigma(A_k)/sqrt(M). Although the phrasing in L.252 is present in literature, it is imprecise and should read "the ensamble standard deviation of the mean differences"

R14) Eq. 18 is corrected, and the phrase "the ensemble standard deviation of the mean difference" is used.

C15) L.255: "... error the ..." is "... error of the ..."
R15) Corrected.

C16) §4.5: this sect. is a 3-line section. For self-contentedness, I suggest to briefly report the Immler check, or to avoid heading this sentence as section.
R16) The description of the consistency check is detailed in the revised manuscript.

Reply to RC2

Thank you very much for your reviewing our manuscript and providing us with valuable comments and suggestions. We reprocessed the IMS-100-GDP because we found some minor issues in the processing data used in the previous manuscript. Furthermore, we applied the updated screening of data following review comments. The number of samples has increased from 57 to 59. However, the main message of the manuscript has unchanged.

Hereafter, Cx represents the referee's comments and Rx represents the reply to Cx.

Major comments

C1) The methods described in sections 2 through 4 are possibly in several cases a bit too brief and too heavily reliant upon the reader going back to and reading a number of previously published papers. Perhaps some key additional aspects need to be documented to ensure broad-scale methodological reproducibility. In particular section 4.5 is too brief given its overall importance in a GRUAN product comparison and should be expanded.

R1) Detailed descriptions are added to the revised manuscript.

C2) For figure 3 and subsequent similar figures it would be useful to describe in the figure 3 caption what the different shapes of the boxes denote? What is a circle, a parallelogram, a diamond, and an oblong?

R2) Explanation of the shapes is added to the captions in the revised manuscript.

C3) It would probably be worth spending some time discussing the very marked seasonality of the rejection rates shown in Figure 10 which is cited but not really discussed. Rejection rates are low in winter but very high in spring and summer. Why is this?

R3) A brief discussion is added to the revised manuscript.

C4) I always hate making this comment because it is immensely impressive for non-native speakers to produce papers in English. However, the paper would be much more readable if you could get a native speaker to edit for clarity. There are numerous places where minor edits would improve the readability and make the messaging stronger.

R4) Thank you very much for your suggestions for edits.

Minor comments

C1) A reader would reasonably ask in the abstract why you mention 99 dual soundings but proceed to

analyze only 57 of these. Can a few words be added to clarify why this 57 subset of the sample were analysed? Something like "Following data quality checks 57 flights were considered of sufficient data quality to produce GDP profiles and this subset is analysed"

R1) We think that this comment is related to the major comment C3. The discussion is added to the revised manuscript.

C2) Line 8 I would delete "with RS92-GDP" as this is already clear from earlier in the sentence

R2) Deleted.

C3) Line 19-20 I would write. "While the RS92-SGP radiosonde has a GDP it was required to seek alternative radiosonde models to use for operational reasons as the payloads often fall within the greater Tokyo metropolitan region and for health and safety reasons use of lighter instrumentation is necessary."

R3) Revised in response to your suggestion.

C4) In lines 21-22 I would suggest making explicitly clear that the RS-11G has already been developed and certified as a GDP

R4) The following sentence is added: "The GDP for RS-11G was developed and certificated in 2019."

C5) Line 56 and not ant (typo)

R5) Corrected.

C6) Should line 105 not be 10' x 10' ? A geoid model at 10 degrees by 10 degrees feels implausibly coarse and the change to 5' by 5' later in paragraph is then huge.

R6) Co-authors at Meisei confirmed that the original geoid model used for iMS-100 is 10 degrees by 10 degrees.

C7) Perhaps in 2.2.4 make clear why no pressure sensor is fitted. Presumably this is to save weight and because the errors in the hydrostatic equation based approximation are considered sufficiently small to justify the omission of such a sensor?

R7) Yes, and the cost reduction is also one of the reasons. The following sentences are added: "Not installing a pressure sensor is to reduce weight and manufacturing costs. However, in recent years, radiosondes without a pressure sensor have tended to be used because the accuracy of the atmospheric pressure derived using hydrostatic equation-based approximation is sufficient (e.g., Nash et al., 2011)."

C8) Lines 205-206 are not possible to follow logically. How did a single payload last 7 months? What

is a logistic regression? Work required here to clarify this sentence please.

R8) Logistic regression analysis is used for probability prediction. Training data is the routine data (twice per day) at Tateno. The beggining of the paragraph is revised as follows: "For screening of ice-contaminated profiles, the probability of icing, $Pr_{ice}$, is derived using logistic regression analysis after variable selection from the length of ISSR, RH and the volume mixing ratio at several levels. The 452 routine observations (twice per day) with a single payload taken from April to November 2018 at Tateno are used as training data." And the reference of the logistic regression is added to the revised manuscript.

C9) Lines 218-219 are again unclear. I think you mean something like: The criteria applied here apply solely to the present study and are not applied to the GDP. However, you also probably need to explain why this is the case. Surely the same processing should be applied to the GDP screening or is there some reason why this would not be possible?

R9) There are no authorized screening methods for the GDPs. This screening method is unique to this study. Users of the data may use their own methods of screening for their interests. The concept of screening with the uncertainties is added to the revised manuscript as follows: "The second screening is based on the uncertainty amounts (Sommer, 2013). This screening is based on the idea that data with uncertainties exceeding the criteria are of questionable reliability and need to be verified individually. "

C10) The paragraph starting line 302 is presumably applicable to all meteorological elements measured and not just temperature. As such its placement here rather than elsewhere – perhaps most logically the discussion section of the paper – feels strange to me. There is already similar text on lines 398-405 so maybe you can just delete this?

R10) This paragraph was deleted in the revised manuscript.

C11) Line 388-389 it is unclear for what parameter this finding applies. I assume pressure but it needs to be stated explicitly.

R11) The sentence will revised as follows: "While there are some cases where with significant differences for pressure are observed in the lower troposphere ($\geq$ 700 hPa) in the consistency check, the mean pressure difference is less than 0.4 hPa."

Reply to RC3

Thank you very much for your reviewing our manuscript and providing us with valuable comments and suggestions. We reprocessed the IMS-100-GDP because we found some minor issues in the processing data used in the previous manuscript. Furthermore, we applied the updated screening of data following your review comment. The number of samples has increased from 57 to 59. However, the main message of the manuscript has unchanged.

Hereafter, the referee's comments are numbered as Cx and Rx represents the reply to Cx.

C1) [line 18] It would be useful to state the weight of the RS-92 sonde so that the relevant lightness of the RS-11G sonde (85g) is clear.

R1) Added.

C2) [line 31] expand first use of 'LC'

R2) Done.

C3) [line 56] change 'ant' for 'and'

R3) Corrected.

C4) [line 80] and Table 2/3 – tables 2 and 3 identifies the uncertainty sources in the temperature and RH products and these include identification of correlated and uncorrelated sources – but this is the only reference I could see in the paper to this separation of uncertainty source classes. Discussion should be included in the main text about this separation and the implication for the expected level of agreement in the intercomparison of multiple flights.

R4) In the revised manuscript, a brief explanation is added for the classification of correlated and uncorrelated uncertainties with examples.

C5) [lines 81 – 101] The section on the RH correction could be clearer in terms of the processing steps made as well as the justification (and robustness) of the assumed values of the key parameters. So, for example three parameter values are given without source information [lines 91-92] and then only two are apparently used in the calculation [lines 93-95].

R5) The description of RH correction processing steps is detailed in the revised manuscript.

C6) [lines 130-132] The parameters in the text are not consistent with the parameters given in Figure 7 and Tpend is undefined.

R6) The explanation of parameters is added to the revised manuscript.

C7) [line 165] missing comma after 'plastic'

R7) We meant "corrugated plastic board" by "plastic cardboard" (i.e., "plastic cardboard" is a one single material name, not two materials). Revised.

C8) [line 195] according to Fig 9 the RH for iMS-100 does decrease in the stratosphere, just not as rapidly as for the RS-92 results.

R8) For RS92, there is little need to consider potential freezing of its humidity sensor since it is heated. On the other hand, the very slow RH decreasing after passage through supercooled layers (red shaded in the figure) and (relatively) high RH values in the stratosphere for iMS-100 are not explained by the hysteresis, but probably due to icing. The sentences are revised as: "For RS92, there is little need to consider potential freezing of its humidity sensor since it is heated. On the other hand, the very slow RH decreasing after passage through a supercooled layer (red shaded in Fig. 10 (b)) and relatively high RH values in the stratosphere for IMS-100-GDP are not explained by the hysteresis, but probably due to contamination and changes in the RH sensor specifications related to icing or freezing during passage through supercooled droplet clouds."

C9) [line 205] should 'logistic regression' be 'logarithmic regression'?

R9) No, 'logistic regression' , which is used for the probability prediction. The reference is added to the revised manuscript.

C10) [line 207] more explanation of the source of this expression would be useful, for example the reason for the altitudes of ST1 and ST2 and why RH and VMR values are both included.

R10) To determine the variables for logistic regression analysis, the variable selection from ISSR parameters and RH and VMR values at several altitudes with decision tree was tested. But describing the detail of the variable selection is not the main point of this article, so only a brief explanation is added. The sentence is revised as: "For screening of ice-contaminated profiles, the probability of icing, $Pr_{ice}$, is derived using logistic regression analysis after variable selection from the length of ISSR, RH and the volume mixing ratio at several levels."

C11) [lines 211 – 220] I had a number of questions over the uncertainty screening section. Firstly, the purpose of the uncertainty screening should be clarified in terms of what data issues it aims to address - as the GDP processing aims to provide detailed uncertainty information this information should, in itself, define how many points would lie outside the uncertainty bounds for a given confidence limit. This is fully expected within a data distribution and outliers should not be eliminated on this basis, particularly if a later assessment looks at the level of agreement between two data sets. Secondly, it is

stated that the coefficients are determined empirically based on a 90% criterion – was this done of the complete data set (and so, by default excludes 10% of the data)? Finally, since the uncertainties for T and RH are significantly different for day-time and night-time flights it would seem sensible for separate screening criteria to be used for the two cases.

R11)

1) The screening described in Sect. 4.1 is based on the idea that data with uncertainties exceeding the criteria (thus, the uncertainty is the outlier) are of questionable reliability and need to be verified individually (so they are classified as "checked" for RS92-GDP.2). In the revised manuscript, a figure for the case which is excluded by screening by LC but adopted in this study is added.

2) The data set used to determine the threshold formula for IMS-100-GDP was not the data set used for the dual-flight intercomparison but for routine (i.e., single) observations from April to November 2018. The distribution of the uncertainties was tabulated per appropriate interval of values (T, RH, and P), and the envelope of data that would not be an outlier was determined by regression. The result is Equation 18. The "ratio criteria is set to 90%" means that if more than 90% of the whole profile has data whose uncertainty does not exceed the threshold, the IMS-100-GDP is adopted. Some examples are shown in the revised manuscript.

3) As you pointed out, the temperature uncertainties for daytime observations become larger than nighttime due to radiation correction. However, since most of the uncertainties, for both daytime, and nighttime, are within the thresholds indicated in Eq. 18, we do not consider it necessary to distinguish between them. On the other hand, we have reconsidered your point about RH uncertainties to use the respective discrimination formulas for daytime and nighttime. As a result, two cases in the daytime were added to the verification.

C12) [line 231] some parameters in this equation are undefines (REGSEE and s)

R12) RESGEE is expanded and revised equation (change s to t).

C13) [line 237] is this criteria correct, or should it say that profiles with more than 10% of abnormal points are excluded?

R13) The latter is what we meant. The sentence is revised as: "Profiles with > 10 % of abnormal data points …"

C14) [line 239] what was the criteria for 'abnormal wind data' in this context?

R14) No screening for wind was done (for both GDPs).

C15) [line 251] should this be the sum over j from 1 to M (as in eq 18) ?

R15) Corrected.

C16) [line 255] 'MVD' is undefined.

R16) Expanded expression for MVD (mean vector difference) is added to the revised manuscript.

C17) [lines 297-298] The evaluation of 'consistent' and 'in agreement' should take into account the effect of the uncertainty screening and correlated/uncorrelated uncertainties (see previous comments)

R17) In this study, the ratio of consistency check results are used for simultaneous data using the total uncertainties of GDPs. Thus, a brief discussion of the factors involved in uncertainty increases/decreases is added to the revised manuscript.

C18) [line 309-310] clarify what is meant by 'differences...are small' – point by point differences or systematic differences over multiple flights? Small compared to expected difference given uncertainties (which would imply uncertainties are over-estimated) or consistent with expected uncertainties? And similarly for 'seasonal variations are large', with possible reference to the earlier point about correlated and uncorrelated uncertainties.

R18) For line 309–310, the systematic differences are discussed, not considering consistency.

C19) [line 322] same comment as for lines 297-298

R19) As mentioned in A17, only factors involved in uncertainty increases/decreases are discussed.

R20) [line 343] is this the standard or expanded uncertainty? Also, the uncertainty on both barometers should be considered when comparing the results.

R20) This is the standard uncertainty. The phrase "for k=1" is added.

C21) [line 365] what does 'small enough' actually mean ? See previous comments on evaluating consistency.

R21) The result and discussion for wind speed and direction consistency are added in the revised manuscript.

C22) [line 376] see previous comments on evaluating consistency.

A22) A brief discussion about consistency with the SKYDEW measurements is added to the revised manuscript. But please note that the uncertainty for the SKYDEW has not been fully evaluated and thus not calculated here.

C23) [lines 378 – 405] the summary should be updated based on the points raised above.

R23) The discussion about consistency for wind is added.

C24) [line 412] GRUAN data product (rather than processing)
R24) Corrected.

C25) [Figure 3] as SEA is an acronym should it be capitalised?
R25) Corrected.

C26) [Figure 7] wonder if the first box should be modified as current content implies that the wind info is derived from a single lat and long value?
R26) "lat0" and "lon0" mean the set of initial latitude and longitude values, not a single value. The main text is revised.

---

## Referee Report (RR1)

**Review of "Comparison of GRUAN Data Products for Meisei iMS-100 and Vaisala RS92 Radiosondes at Tateno, Japan" by Shunsuke Hoshino et al.**

The authors have undertaken a set of revisions that respond to the majority of the comments left by the three reviewers to the public review version. It is perhaps unfortunate that a comparison of the uncertainty ranges is not given (as suggested by reviewer 1). Reviewer 3 also has a legitimate concern that the screening may impact the comparison in important ways when, as the authors have clarified, the screening is unique to this comparison and will not be applied operationally.

Overall, the work remains publishable. I suggest at a minimum the authors attend to the points and suggestions made below.

**Major comments**

- 1. It may be valuable in the discussion around line 405-409 to discuss what the comparisons to both satellite data and the new RS41 product imply about whether this temperature difference may arise in the current RS92 processing.
- 2. It would be worthwhile considering whether a reason for the difference in RH behaviour in the presence of sharp gradients as discussed at the end of page 15 could be given. I assume it arises because of the difference between having a heated sensor or a passive sensor. I would furthermore hypothesise that the effect would be more marked in going from high to low humidity than from low to high humidity if this were the case. It is well known that passive sensors have issues of residual wetting on exit from cloud tops such that there is an asymmetry in the effect.

**Minor comments**

- 1. Line 7 I would say 0.5 K cooler rather than 0.5 K lower
- 2. Line 106 hese -> These at start of the sentence
- 3. Line 109 called as the -> termed the
- 4. Line 146 change to 'near Japan is one of the regions with large differences'
- 5. Line 184 it is unclear what you mean by supporting latitude and longitude. I assume you mean requiring latitude and longitude information or similar?
- 6. Line 191 Each of these components [...]
- 7. Line 198 convert -> converts
- 8. Line 205 which of the two correction models is [...]
- 9. Line 248 uncertainty amounts -> quantified uncertainty estimates
- 10. Lines 392-394 do you not need to make clear that the consistency ranks correspond to satisfying k<1, k<2, k<3 and k>3 respectively?
- 11. Line 428 change 'of' to 'the'
- 12. Line 432 reason for these difference could be [...]
- 13. Line 473 different -> difference
- 14. Several of the figures have very small font size. Where possible increasing the font size would increase the figure readability.

---

## Author Response (AR2)

**Reply to report #1 by anonymous referee 2**

Thank you very much for your reviewing our manuscript and providing us with valuable comments and suggestions. Hereafter, Cx represents the referee's comments and Rx represents the reply to Cx.

**Major comments**

C1) It may be valuable in the discussion around line 405-409 to discuss what the comparisons to both satellite data and the new RS41 product imply about whether this temperature difference may arise in the current RS92 processing.

R1) Thank you for the suggestion of the verification using third data (satellite or other types of radiosonde). But, the GNSS-RO temperature data are very limited and cannot be used for additional verification. And we have not conducted iMS-100 vs RS92 vs RS41 comparison flights. So this issue will be discussed according to the comparisons between RS92 vs RS41 and iMS-100 vs RS41 later.

So we add sentences like the below:

"Further discussion about the contributions of different radiation heating correction methods to the temperature difference needs other observation data like satellites or other types of radiosonde, like RS41. But GNSS-RO-based temperature data is very limited and no comparative observations have been made at Tateno between three sondes (iMS-100, RS92, and RS41). Therefore, additional discussion is expected after the results of comparisons between iMS-100 vs RS41 and RS92 vs RS41 are published."

C2) It would be worthwhile considering whether a reason for the difference in RH behaviour in the presence of sharp gradients as discussed at the end of page 15 could be given. I assume it arises because of the difference between having a heated sensor or a passive sensor. I would furthermore hypothesise that the effect would be more marked in going from high to low humidity than from low to high humidity if this were the case. It is well known that passive sensors have issues of residual wetting on exit from cloud tops such that there is an asymmetry in the effect.

R2) As you mentioned, the iMS-100's RH sensor shows residual wetting (termed as hysteresis, described in Section 2.2.2), but RS92's RH sensor does not show hysteresis because of heating. I think you suggest this point must be emphasized. So the sentences like the below are added: "As described in Section 2.2.2, the iMS-100's RH sensor has hysteresis with the large time constant, but RS92's RH sensor is heated and its hysteresis is negligible. This difference in characteristics of RH sensor could cause the large difference, especially in rapid decreasing RH case."

Minor comments

C1) Line 7 I would say 0.5 K cooler rather than 0.5 K lower

R1) Rephrased.

C2) Line 106 hese -> These at start of the sentence

R2) This is typo. Corrected.

C3) Line 109 called as the -> termed the R3) Rephrased.

C4) Line 146 change to 'near Japan is one of the regions with large differences' R4) Rephrased.

C5) Line 184 it is unclear what you mean by supporting latitude and longitude. I assume you mean requiring latitude and longitude information or similar?

R5) Rephrased as "the initial wind speed wspeed0 and direction wdir0 are derived as motion vectors from longitude ( $\lambda$ ; lon0 in Fig. 8) and latitude ( $\phi$ ; lat0 in Fig. 8) based on GPS positioning for IMS-100-GDP"

C6) Line 191 Each of these components [...]R6) Rephrased.

C7) Line 198 convert -> convertsR7) Corrected.

C8) Line 205 which of the two correction models is [...]R8) Rephrased.

C9) Line 248 uncertainty amounts -> quantified uncertainty estimatesR9) Rephrased.

C10) Lines 392-394 do you not need to make clear that the consistency ranks correspond to satisfying k<1, k<2, k<3 and k>3 respectively?

R10) I understand you pointed out that consistency ranks are k<1, 1<=k<2, 2<=k<3 and k>=3 in Table 8, not as 1, 2, 3 and 4. So I rephrased the text as so.

C11) Line 428 change 'of' to 'the' R11) Corrected.

C12) Line 432 reason for these difference could be [...] R12) Rephrased.

C13) Line 473 different -> difference R13) Corrected.

C14) Several of the figures have very small font size. Where possible increasing the font size would increase the figure readability.

R14) We increase the font size in Figs. 4, 12, 13, 15, 16, and 17. Fig. 1 is enlarged.

Reply to report #2 by anonymous referee 1

Thank you very much for your reviewing our manuscript and providing us with valuable comments and suggestions. Hereafter, Cx represents the referee's comments and Rx represents the reply to Cx.

C1) Line 91: Uncertainties - > error

R1) I'm sorry but I cannot find these points in the change-tracking file.

C2) Line 146-7: are -> is

R2) I'm sorry but I cannot find these points in the change-tracking file.

C3) Lines 394-397 are a quotation from Immler et al. 2010 and should be between quotes or, better, appropriately rephrased.

- Although Immler et al. words are a standard in the atmospheric measurement community, there is a confounding use of the term uncertainty as it is defined in metrology and of the concept of equality of measurements (m1\_=m\_2) as is specified in statistics and probability.
- According to standard terminology:
  - "uncertainty" is a spread parameter, in our case u\_1 or u\_2, related to the measurement error. Hence instead of "uncertainty is normally distributed", the sentence "measurement error is normally distributed" should be used.
  - "Assuming the hypothesis that m\_1=m\_2 is true" should be rephrased as "Assuming that the two measurements m\_1 and m\_2 have the same expectation (or mean)."
- To see this, note that the sentence of Immler et al., "the probability that |m\_1m\_2|>k(u\_1^2 + u\_2^2)^0.5 occurs only by chance, is roughly 4.5% for k = 2 and 0.27% for k = 3" is correct under the assumptions that the measurements m\_1 and m\_2 have independent errors with measurement uncertainties (or standard errors) u\_1 and u\_2, and the difference m\_1-m\_2 has a normal distribution.
- In fact, "the hypothesis that m\_1=m\_2 is true" is a confounding statement because, due to the normal distribution, the event m\_1=m\_2 has a zero probability of happening by chance. Also, it cannot be interpreted as "equality in distribution" or "equality of the two distributions" because u\_1 and u\_2 may differ.

R3) Thank you for your explanation. We revise this part as follows:

"Under the assumptions that the measurements m\_1 and m\_2 have independent errors with measurement uncertainties (or standard errors) u\_1 and u\_2, and the difference m\_1-m\_2 has a normal distribution, the probability that ... "